# Contrasting source contributions of Arctic black carbon to atmospheric concentrations, deposition flux, and atmospheric and snow radiative effects

Hitoshi Matsui[1], Tatsuhiro Mori[2,3], Sho Ohata[4,5], Nobuhiro Moteki[2], Naga Oshima[6], Kumiko Goto-Azuma[7,8], Makoto Koike[2], and Yutaka Kondo[7]

[1] Graduate School of Environmental Studies, Nagoya University, Nagoya, Japan
[2] Graduate School of Science, University of Tokyo, Tokyo, Japan
[3] Faculty of Science and Technology, Keio University, Yokohama, Japan
[4] Institute for Space–Earth Environmental Research, Nagoya University, Nagoya, Japan
[5] Institute for Advanced Research, Nagoya University, Nagoya, Japan
[6] Meteorological Research Institute, Tsukuba, Japan
[7] National Institute of Polar Research, Tachikawa, Japan
[8] The Graduate University for Advanced Studies, Hayama, Japan

*Correspondence to*: Hitoshi Matsui (matsui@nagoya-u.jp)

**Abstract.** Black carbon (BC) particles in the Arctic contribute to rapid warming of the Arctic by heating the atmosphere and snow and ice surfaces. Understanding the source contributions to Arctic BC is therefore important, but they are not well understood, especially those for atmospheric and snow radiative effects. Here we estimate simultaneously the source contributions of Arctic BC to near-surface and vertically integrated atmospheric BC mass concentrations ($M_{BC\_SRF}$ and $M_{BC\_COL}$), BC deposition flux ($M_{BC\_DEP}$), and BC radiative effects at the top of the atmosphere and snow surface ($RE_{BC\_TOA}$ and $RE_{BC\_SNOW}$), and show that the source contributions to these five variables are highly different. In our estimates, Siberia makes the largest contribution to $M_{BC\_SRF}$, $M_{BC\_DEP}$, and $RE_{BC\_SNOW}$ in the Arctic (defined as >70°N), accounting for 70%, 53%, and 41%, respectively. In contrast, Asia's contributions to $M_{BC\_COL}$ and $RE_{BC\_TOA}$ are largest, accounting for 37% and 43%, respectively. In addition, the contributions of biomass burning sources are larger (29−35%) to $M_{BC\_DEP}$, $RE_{BC\_TOA}$, and $RE_{BC\_SNOW}$, which are highest from late spring to summer, and smaller (5.9−17%) to $M_{BC\_SRF}$ and $M_{BC\_COL}$, whose concentrations are highest from winter to spring. These differences in source contributions to these five variables are due to seasonal variations in BC emission, transport, and removal processes and solar radiation, as well as to differences in radiative effect efficiency (radiative effect per unit BC mass) among sources. Radiative effect efficiency varies by a factor of up to 4 among sources (1471−5326 W g$^{-1}$) depending on lifetimes, mixing states, and heights of BC and seasonal variations of emissions and solar radiation. As a result, source contributions to radiative effects and mass concentrations (i.e., $RE_{BC\_TOA}$ and $M_{BC\_COL}$, respectively) are substantially different. The results of this study demonstrate the importance of considering differences in the source contributions of Arctic BC among mass concentrations, deposition, and atmospheric and snow radiative effects for accurate understanding of Arctic BC and its climate impacts.

## 1 Introduction

Black carbon (BC) aerosols, emitted into the atmosphere by incomplete combustion of fossil fuels, biofuels, and biomass, heat the atmosphere and modulate the Earth's radiation budget by efficiently absorbing solar radiation (e.g., Bond et al., 2013; IPCC, 2021). This heating by BC also changes the vertical stability of the atmosphere and the distribution of clouds, which in turn modulates the radiation budget (e.g., Koch and Del Genio, 2010; Smith et al., 2018; Stjern et al., 2017). In addition, when BC is transported and deposited in regions where snow and ice are present, such as in the Arctic region, it lowers the albedo of snow and ice surfaces and accelerates snow and ice melting (e.g., Flanner et al., 2007; Hadley and Kirchstetter, 2012; Hansen and Nazarenko, 2004). Arctic warming is progressing about twice as fast as global warming, and BC in the Arctic may contribute to the acceleration of Arctic warming through heating of the atmosphere and heating and melting of snow and ice

(e.g., Serreze and Barry, 2011). However, there are large uncertainties in simulations of atmospheric BC concentrations in the Arctic, which vary by one or two orders of magnitude among existing models (e.g., Eckhardt et al., 2015; Shindell et al., 2008).

Atmospheric BC mass ($M_{BC}$) concentrations in the Arctic show distinct seasonal variation, being high in winter and spring and low in summer near the surface (e.g., Sharma et al., 2013, 2019; Sinha et al., 2017). In winter and spring, BC emitted from high latitudes such as from Siberia and Europe is transported via the lower troposphere to lower altitudes in the Arctic, whereas anthropogenic BC emitted from mid-latitudes such as from Asia is transported long distances via the middle and upper troposphere and reaches higher altitudes in the Arctic (e.g., Stohl, 2006; Matsui et al., 2011a). In summer, when precipitation in the mid- and high latitudes increases, biomass burning in and near the Arctic (Siberia and Alaska) is considered to be the dominant source of atmospheric BC in the Arctic (e.g., Ikeda et al., 2017; Sharma et al., 2013). Unlike atmospheric BC, the rain rate in the Arctic varies seasonally, with a lower rain rate in winter and spring and higher in summer (e.g., Mori et al., 2020; Shen et al., 2017). Thus, the BC deposition flux has been reported to have seasonal variations with maximum fluxes in summer (or less clear seasonal variations than surface atmospheric BC) (Mori et al., 2020, 2021). Furthermore, heating of the atmosphere and snow surfaces in the Arctic by BC is strongly dependent on solar radiation, which is largest in summer (and zero in winter because of the polar night). Therefore, atmospheric concentrations and deposition of BC from spring to fall are important for estimating the heating of the atmosphere and snow surface by BC in the Arctic. Given the seasonal variations in BC emission, transport, and removal processes, as well as in solar radiation, the source contributions to the following five BC variables and their seasonal variations are expected to differ significantly in the Arctic: 1) near-surface atmospheric BC mass concentration ($M_{BC\_SRF}$), 2) vertically integrated atmospheric BC mass concentration ($M_{BC\_COL}$), 3) BC deposition flux ($M_{BC\_DEP}$), 4) BC radiative effect at the top of the atmosphere (TOA) ($RE_{BC\_TOA}$), and 5) BC radiative effect at the snow surface ($RE_{BC\_SNOW}$).

Many previous studies have estimated source contributions to BC in the Arctic. Most of them have focused on $M_{BC\_SRF}$ and $M_{BC\_COL}$, showing that BC in the lower troposphere of the Arctic is mainly transported from high-latitude sources such as Europe, Siberia, and North America, whereas low-latitude sources such as Asia are important contributors to BC in the middle and upper troposphere (e.g., Bourgeois and Bay, 2011; Huang, et al., 2010; Qi et al., 2017; Ren et al., 2020; Sharma et al., 2013; Sobhani et al., 2018; Xu et al., 2017; Zhu et al., 2020). Other studies have estimated the source contributions to $M_{BC\_DEP}$ in the Arctic as well as those to $M_{BC\_SRF}$ and $M_{BC\_COL}$ (Ikeda et al., 2017; Qi and Wang, 2019; Wang et al., 2011). In contrast, few studies have estimated the source contributions to the radiative effects of BC ($RE_{BC\_TOA}$ and $RE_{BC\_SNOW}$) in the Arctic. As far as we know, only Wang et al. (2014) have estimated the source contributions to $M_{BC\_SRF}$, $M_{BC\_COL}$, $M_{BC\_DEP}$, and $RE_{BC\_TOA}$ in the Arctic. They found strong seasonal variations in source contributions and showed the importance of high-latitude sources. However, their model simulations underestimated observed BC mass concentrations at the surface and in the lower troposphere by about one order of magnitude.

To our knowledge, no study has estimated the source contributions to all five of the BC variables described above ($M_{BC\_SRF}$, $M_{BC\_COL}$, $M_{BC\_DEP}$, $RE_{BC\_TOA}$, and $RE_{BC\_SNOW}$) simultaneously. In addition, although BC emitted from each source may have different microphysical properties (e.g., mixing state) and radiative effect efficiency (radiative effect per unit light BC absorption or per unit BC mass) in the Arctic, these differences among emission sources are not well understood.

In our previous studies, we have developed a global two-dimensional sectional aerosol model, the Community Atmosphere Model with the Aerosol Two-dimensional bin module for foRmation and Aging Simulation (CAM-ATRAS), that resolves aerosol particle size and the BC mixing state in detail (Matsui, 2017; Matsui and Mahowald, 2017). We have also shown that simulations conducted with this model can reproduce realistically global distributions of $M_{BC}$ observed by surface and aircraft measurements (e.g., Matsui and Mahowald, 2017; Liu and Matsui, 2021b). In this study, we use CAM-ATRAS to estimate the source contributions to BC in the Arctic from 26 sources (13 source regions × 2 source types (anthropogenic and biomass burning)) and show that source contributions to Arctic BC are substantially different among the five BC variables: $M_{BC\_SRF}$, $M_{BC\_COL}$, $M_{BC\_DEP}$, $RE_{BC\_TOA}$, and $RE_{BC\_SNOW}$. We also show in this study that the radiative effect efficiency of atmospheric

BC in the Arctic differs significantly among emission sources, and that these differences contribute to the different source contributions to atmospheric concentrations and radiative effects (i.e., $M_{BC\_COL}$ and $RE_{BC\_TOA}$). Abbreviations for BC used in this study are summarized in Table 1.

## 2 Method

### 2.1 Global climate-aerosol model CAM-ATRAS

We used the Community Atmosphere Model version 5 (CAM5) (Neale et al., 2010) and the Community Land Model version 4 (CLM4) (Oleson et al., 2010) in Community Earth System Model version 1.2.0 (Hurrell et al., 2013). In our previous studies, we implemented our aerosol model ATRAS into CAM5 (Matsui, 2017; Matsui and Mahowald, 2017). In CAM-ATRAS, which considers seven aerosol species (sulfate, nitrate, ammonium, dust, sea salt, organic aerosol, and BC), aerosol particles with dry diameters from 1 to 10,000 nm are classified into 12 particle size bins, and for fine particles (5 particle size bins from 40 to

1250 nm), eight BC mixing state bins are used for each size bin. Based on the mass ratio of BC to total dry aerosol (fBC), the BC mixing states are classified into pure BC (fBC = 0.99−1.0), BC-free particles (fBC < 0.0001), and six different internally mixed BC particles (fBCs of 0.0001−0.1, 0.1−0.2, 0.2−0.5, 0.5−0.8, 0.8−0.9, and 0.9−0.99). Overall, 47 particle size and mixing state bins are used to represent aerosols. CAM-ATRAS calculates the following aerosol processes for the full two-dimensional bins (47 bins): new particle formation (Matsui et al., 2011b, 2013a); condensation of sulfate, nitrate, and organic

aerosols (Matsui et al., 2014a, 2014b); coagulation (Matsui et al., 2013b); activation into cloud droplets (Abdul-Razzak and Ghan, 2000, 2002); aqueous-phase chemistry (Tie et al., 2001); and dry and wet deposition (Liu et al., 2012; Zender et al., 2003). Changes in particle size and mixing state by condensation, coagulation, and aqueous-phase formation are calculated for all the 47 bins, and bin shifting by these processes is calculated by a two moment (mass and number) advection scheme (Simmel and Wurzler, 2006) for particle size bins and the moving center approach (Jacobson, 1997) for mixing state bins

(Matsui, 2017). Optical properties and cloud condensation nuclei (CCN) properties are calculated theoretically (Bohren and Huffman, 1998; Petters and Kreidenweis, 2007) using the particle size and chemical composition of each two-dimensional bin, and aerosol-radiation (Iacono et al., 2000) and aerosol-cloud interactions (Morrison and Gettelman, 2008) are estimated based on these properties (Matsui, 2017). For optical properties, we assumed the core/shell treatment for internally mixed BC in the fine particle size bins (40−1250 nm in diameter) and the well-mixed treatment for the other particles (for pure BC and BC-free

particles in the fine particle size bins and for all particles larger than 1250 nm or smaller than 40 nm) (Matsui, 2017). CAM-ATRAS uses look-up tables of optical parameters (extinction coefficient, single scattering albedo, and asymmetry factor) calculated based on the codes for homogeneous and coated spheres (Appendices A and B in Bohren and Huffman (1998)). The core/shell treatment could underestimate the mass absorption cross section of BC ($MAC_{BC}$) for large particles (Forestieri et al., 2018), but as shown by Matsui et al. (2018b), the enhancement of BC absorption by the core/shell treatment is

comparable to that by other mixing state assumptions such as the dynamic effective medium approximation (Chylek et al., 1984; Jacobson, 2006) and the Bruggeman mixing rule (Jacobson, 2006).

   Model simulations by CAM-ATRAS have been evaluated against various surface, aircraft, and satellite observations for mass concentrations of each aerosol species, number concentrations, size distributions, and optical properties (Gliβ et al., 2021; Kawai et al., 2021; Liu and Matsui, 2021a; Matsui and Mahowald, 2017; Matsui et al., 2018a; Matsui and Moteki, 2020; Sand

et al., 2021). Mass concentrations, mixing states, and vertical profiles of BC have been validated (Matsui et al., 2018b; Matsui, 2020; Moteki et al., 2019; Ohata et al., 2021a). We have also improved the model representation of activation processes in liquid clouds and of removal processes in cumulus and mixed-phase clouds, thereby greatly improving the reproducibility of BC observations in the upper troposphere in the tropics and in the middle and lower troposphere in the Arctic (Liu and Matsui, 2021b; Matsui and Liu, 2021). In Liu and Matsui (2021b), we separately represented activated and non-activated aerosols in

convective clouds and introduced gradual activation processes of aerosols during upward transport. This representation allows

consistent calculations of the transport, activation, and removal processes of aerosols in convective clouds. We also introduced the reduction in precipitation removal efficiency of aerosols in mixed-phase clouds by the Wegener-Bergeron-Findeisen process (Liu and Matsui, 2021b), and following Cozic et al. (2007), we represented precipitation removal efficiency as a function of the ice mass fraction in mixed-phase clouds.

## 2.2 Tag-tracer method

In addition to the sum of BC from all source regions and types (hereafter referred to as ALL BC), CAM-ATRAS considers two tracer BC variables. The tracer BC variables are considered for all 47 particle-size and mixing-state bins. Transport, aging, and removal processes and related changes in particle sizes and mixing states of the tracer BC variables are calculated explicitly and in the same way as those of the ALL BC. In the original CAM-ATRAS, these two tracer BC variables are used to calculate anthropogenic (fossil fuel + biofuel) and biomass burning BC from all source regions. In this study, we used these tracer BC variables to calculate anthropogenic and biomass burning BC emitted from each of 13 regions (Fig. 1): Europe (EUR), Siberia (SIB), Greenland (GL), North America north of 50°N (NAM (>50°N)), North America south of 50°N (NAM (<50°N)), Central Asia (CAS) 1−4, East Asia (EAS) 1−2, Southeast Asia (SAS), and Others. By performing 13 simulations focusing on anthropogenic and biomass burning BC emitted from each source region, the emission, transport, aging, and removal processes and optical and CCN properties of BC from all 26 sources (i.e., 13 regions × 2 types) are calculated separately using the 47 bins for each emission source.

By using these tag-tracer BC variables, source contributions to BC in the Arctic (defined as >70°N in this study) were estimated for the five BC variables: $M_{BC\_SRF}$, $M_{BC\_COL}$, $M_{BC\_DEP}$, $RE_{BC\_TOA}$, and $RE_{BC\_SNOW}$. For $M_{BC\_SRF}$ and $M_{BC\_DEP}$, ALL BC and the sum of all BC tags from the 26 sources agree within 0.40% for the global and Arctic averages (Fig. S1). Regarding the spatial distributions of $M_{BC\_SRF}$ and $M_{BC\_DEP}$, ALL BC and the sum of all BC tags show good agreement in almost all grids globally. For $M_{BC\_COL}$, ALL BC and the sum of all BC tags agree within 3.0% for the global and Arctic averages, and within 10% for all grids in the Arctic (Fig. S1).

$RE_{BC\_TOA}$ for each source is estimated from the difference between when all BC is considered and when BC from the target source is excluded from all BC. Three radiative transfer calculations (considering ALL BC, excluding anthropogenic BC in the target area from ALL BC, and excluding biomass burning BC in the target area from ALL BC) were performed for each simulation to estimate instantaneous BC radiative effects from the target source. $RE_{BC\_TOA}$ for ALL BC and that for the sum of all BC tags agree within 10% for global and Arctic averages and for almost all grids in the Arctic (Fig. S1).

$RE_{BC\_SNOW}$ is calculated by the Snow, Ice, and Aerosol Radiative (SNICAR) model in CLM4 (Flanner and Zender, 2005; Oleson et al., 2010). Similar to $RE_{BC\_TOA}$, we tried to estimate $RE_{BC\_SNOW}$ for each source from the difference between $RE_{BC\_SNOW}$ when all BC is considered and when BC of the target source is excluded from all BC. However, using this method, the difference between ALL BC and the sum of all BC tags is more than 10% in many grids, and it is 20% and 5.8% for the global and Arctic averages, respectively (Fig. S1).

Given these results, this online calculation is not used in this study; instead, $RE_{BC\_SNOW}$ for each source ($RE_{BC\_SNOW,i,m,s}$) is estimated offline using Eq. (1):

$$RE_{BC\_SNOW,i,m,s} = RE_{BC\_SNOW,i,m,ALL} \times \frac{M_{BC\_DEP,i,m,s} + M_{BC\_DEP,i,m-1,s}}{\sum_s (M_{BC\_DEP,i,m,s} + M_{BC\_DEP,i,m-1,s})}, \qquad (1)$$

where $i$, $m$, and $s$ denote a horizontal grid, month, and emission source, respectively; $RE_{BC\_SNOW,i,m,ALL}$ denotes $RE_{BC\_SNOW}$ in horizontal grid $i$ and month $m$ when considering BC from all sources (monthly mean); $M_{BC\_DEP,i,m,s}$ denotes BC deposition flux in horizontal grid $i$, month $m$, and emission source $s$ (monthly mean). Thus, Eq. (1) calculates $RE_{BC\_SNOW,i,m,s}$ from $RE_{BC\_SNOW,i,m,ALL}$ by weighting the contribution of each emission source $s$ to the total BC deposition flux. This offline method assumes that the source contributions to $RE_{BC\_SNOW}$ in a given month are determined by the source contributions to the BC deposition flux in that month and the previous month (two months). In reality, BC older than two months may contribute to

snow surface heating to some extent, and the heating may also depend on the timing and amount of snowfall and variations of snow grain size. Note that varying the weighting period of the deposition flux from 1 to 3 months does not change the estimates of the source contributions (Fig. S2). In addition, the source contributions calculated by the offline calculation (Eq. 1) and those estimated by the online calculation agree well except for Siberia, North America (>50°N), and Central Asia: the offline calculation shows a larger contribution from North America (>50°N) and a smaller contribution from Siberia and Central Asia than the online calculation (Fig. S2). Considering these results, in this study we mainly use the source contributions to $RE_{BC\_SNOW}$ estimated by the offline calculation (Eq. 1).

## 2.3 Simulation setups

Model simulations were performed for eight years, 2008−2015, and the results for the latter seven years, 2009−2015, were used for analysis. As described in Sect. 2.2, 13 simulations were performed using the tag-tracer variables for BC emitted from each of the 13 regions shown in Fig. 1. The horizontal resolution was 1.9º latitude ×2.5º longitude, and the number of vertical layers was 30 (~40 km). All simulations in this study were nudged by the Modern-Era Retrospective analysis for Research and Applications version 2 (MERRA2) for wind speed and direction and temperature in the free troposphere (<800 hPa). Emission data were taken from monthly anthropogenic emissions based on the Community Emissions Data System (Hoesly et al., 2018) and from daily biomass burning emissions based on the Global Fire Emissions Database (GFED) version 4.1 (van der Werf et al., 2017). Although some recent studies have suggested that biomass burning emissions are underestimated (e.g., Reddington et al., 2016; Mallet et al., 2021), the GFED version 4.1 data were used directly in this study. Dust and sea salt emissions were calculated online (Mårtensson et al., 2003; Monahan et al., 1986; Zender et al., 2003). Similar to Matsui et al. (2018b), anthropogenic and biomass burning emissions were assumed to have number median diameters of 70 nm and 100 nm, respectively (standard deviation 1.8). Given the large uncertainty in the assumption of BC mixing states in emissions (Matsui, 2020), we assumed BC is emitted as pure BC and the other species as BC-free particles (Matsui et al., 2018b). In reality, the mixing state of emitted BC particles depends on the types of sources. Matsui et al. (2018b) made a simulation assuming that 50% of BC mass is emitted as pure BC and the other 50% of BC as internally mixed BC with the shell (organic aerosol) to core (BC) diameter ratio of 1.1 for fossil fuel sources and 1.4 for biofuel and biomass burning sources and showed that global mean $RE_{BC\_TOA}$ in this simulation is about 10% larger than that in the simulation assuming pure BC for all BC emissions.

## 2.4 Observation data

$M_{BC\_SRF}$ was observed by a continuous soot monitoring system at Barrow (71.3°N, 156.6°W), Ny-Ålesund (78.9°N, 11.9°E), Alert (82.5°N, 62.5°W), and Pallas (68.0°N, 24.1°E) (Ohata et al., 2021b). At Barrow and Ny-Ålesund, $M_{BC\_DEP}$ was also observed by a single-particle soot photometer in 2013−2017 (Mori et al., 2020, 2021). We used these surface observation data of $M_{BC\_SRF}$ during 2009−2015 and $M_{BC\_DEP}$ during 2013−2017 to evaluate simulated $M_{BC\_SRF}$ and $M_{BC\_DEP}$ (2009−2015) in the Arctic. We also used observations of $M_{BC}$ in surface snow and the total column of snowpack in Finland (March 2013), Alaska (March 2012−2015), Siberia (March 2013 and April 2015), Greenland (June−July 2012, July−August 2013, July−August 2014, May 2015, and May 2016), and Ny-Ålesund (April 2013) (Mori et al., 2019). In addition, we used aircraft $M_{BC}$ observation data at high latitudes in the Northern Hemisphere during the High-performance Instrumented Airborne Platform for Environmental Research (HIAPER) Pole-to-Pole Observations (HIPPO) campaigns in 2009−2011 (Schwarz et al., 2013; Wofsy et al., 2011), the Arctic Research of the Composition of the Troposphere from Aircraft and Satellites (ARCTAS) campaigns in April and July 2008 (Kondo et al., 2011; Matsui et al., 2011a, 2011c), and the Polar Airborne Measurements and Arctic Regional Climate Model simulation Project (PAMARCMiP) campaign in March−April 2018 (Ohata et al., 2021a). Global Precipitation Climatology Project monthly data (https://psl.noaa.gov/data/gridded/data.gpcp.html) were used to evaluate precipitation amounts in the Arctic.

# 3 Results

## 3.1 Comparisons with observed BC in the Arctic

Model simulations generally reproduce the observed seasonal variations of $M_{BC\_SRF}$ (maximum in winter and minimum in summer) well at Barrow, Ny-Ålesund, and Alert (Figs. 2a−c). The simulated/observed ratios of annual-mean $M_{BC\_SRF}$ are 0.61 at Barrow (25 ng m$^{-3}$ in observations and 15 ng m$^{-3}$ in simulations), 1.5 at Ny-Ålesund (13 ng m$^{-3}$ in observations and 18 ng m$^{-3}$ in simulations), 0.46 at Alert (20 ng m$^{-3}$ in observations and 9.3 ng m$^{-3}$ in simulations), and 1.1 at Pallas (29 ng m$^{-3}$ in observations and 31 ng m$^{-3}$ in simulations); thus, observations and model simulations agree reasonably well at all sites (Figs. 2a−d). At Barrow, simulated $M_{BC\_SRF}$ is underestimated from February to April, but agrees with observed $M_{BC\_SRF}$ within a factor of 2 in the other months. At Alert, simulated $M_{BC\_SRF}$ is also underestimated in late winter and spring, but agrees with observations within a factor of 2 except in February−May. At Ny-Ålesund and Pallas, the observed and simulated $M_{BC\_SRF}$ agree within a factor of 2, except in January and August−November at Ny-Ålesund and in November at Pallas.

Observed $M_{BC\_DEP}$ (by wet deposition) shows seasonal variation with a maximum in summer at Barrow and a minimum in summer at Ny-Ålesund (Figs. 3a and b), reflecting seasonal difference of precipitation between the two sites (Mori et al., 2020, 2021). At Barrow, $M_{BC\_DEP}$ is overestimated especially in August, but observed and simulated $M_{BC\_DEP}$ agree within a factor of 2 in 7 out of 12 months. At Ny-Ålesund, simulated $M_{BC\_DEP}$ is also overestimated in summer, but observed and simulated $M_{BC\_DEP}$ agree within a factor of 2 in 6 out of 12 months. The simulated/observed ratio of annual-mean $M_{BC\_DEP}$ is 2.3 at Barrow and 1.5 at Ny-Ålesund. Note that model simulations generally reproduce observed precipitation and its seasonal variations in the Arctic (Figs. S3 and S4).

The vertical profiles of $M_{BC}$ in the Arctic during the HIPPO campaigns generally show good agreement between observations and model simulations (Figs. 4a−e), except in August (HIPPO5). Liu and Matsui (2021b) greatly improved the agreement of the simulated vertical profiles of $M_{BC}$ with observations by improving aerosol removal processes for cumulus and mixed-phase clouds. The level of agreement of vertical profiles of $M_{BC}$ with observations in this study is similar to that in Liu and Matsui (2021b) for the HIPPO campaigns. The simulations overestimate observed $M_{BC}$ in summer (especially in August) both at Barrow and in the HIPPO5 campaign (Figs 2a, 3a, and 4e). Model simulations might overestimate BC emissions from biomass burning sources in and around Alaska in summer because their contributions to Arctic BC are large in summer (Sect. 3.3).

We also compare our model simulations with aircraft observations in the ARCTAS and PAMARCMiP campaigns (Figs. 4f−h), although the years of observations and model simulations are not the same. Our model-simulated $M_{BC}$ levels (~10 ng kg$^{-1}$) during the spring season in the European Arctic (~80°N) are generally consistent with the observed $M_{BC}$ in the PAMARCMiP campaign (Fig. 4h). The model simulations underestimate $M_{BC}$ in the ARCTAS campaign (Figs. 4f and g) because it is higher than $M_{BC}$ in the HIPPO and PAMARCMiP campaigns. This might reflect the high activity of biomass burning and the resulting high emissions of BC in 2008, when the ARCTAS campaigns were conducted (Ohata et al., 2021a).

Simulated $M_{BC}$ in snow tends to be about a factor of 2−3 higher than observed $M_{BC}$ in snow in Finland, Alaska, Siberia, and Greenland (Fig. S5). However, the simulated $M_{BC}$ in snow agrees with the observations within a factor of 10 at almost all snow sampling sites (Fig. 5a). In addition, the model simulations generally reproduce the observed features with higher $M_{BC}$ in snow over Finland and Siberia and lower over Alaska and Greenland. The simulated $M_{BC}$ in snow has a spatial distribution with higher concentrations in the Siberian side of the Arctic and lower concentrations in the North American side of the Arctic (Fig. 5b), which is consistent with the results of previous studies (e.g., Flanner et al, 2007).

There are uncertainties in comparisons between observations and model simulations. For example, observation data (e.g., aircraft and snow BC data) and model simulation outputs have different spatial and temporal scales. Observed data are for a specific location and time, with time scales of minutes (aircraft observations) to days (snow observations), whereas in comparisons with aircraft observations, we used monthly model outputs for a specific region (e.g., 60−80°N and 140−170°W

for HIPPO) and in comparisons with snow BC, we used monthly averaged model outputs over a horizontal grid of about 200 km. Observations suggest that snow BC concentrations vary widely over fine spatial and temporal scales, but model outputs do not fully resolve this variability (Fig. 5a). These uncertainties in comparisons between observations and models are seen not only in this study but in all studies using both observations and model simulations (e.g., Schutgens et al., 2017). Despite these uncertainties in observation-model comparisons, the results obtained in this study are comparable to or better than those obtained by previous studies in terms of the reproducibility of BC observations in the Arctic.

## 3.2 Spatial distribution of BC

In the Northern Hemisphere, $M_{BC\_SRF}$, $M_{BC\_COL}$, $M_{BC\_DEP}$, and $RE_{BC\_TOA}$ are largest in East Asia and Central Africa, where $RE_{BC\_TOA}$ exceeds 2 W m$^{-2}$ (Figs. 6a–d). Global averages of $M_{BC\_SRF}$, $M_{BC\_COL}$, $M_{BC\_DEP}$, and $RE_{BC\_TOA}$ are 0.14 µg m$^{-3}$, 0.15 Tg, 9.6 Tg y$^{-1}$, and 0.40 W m$^{-2}$, respectively, and Arctic (>70°N) averages are 0.020 µg m$^{-3}$, 0.0016 Tg, 0.052 Tg y$^{-1}$, and 0.31 W m$^{-2}$, respectively. The atmospheric lifetime of BC (ratio of $M_{BC\_DEP}$ to atmospheric BC burden) is estimated to be 5.6 days for the global average and 11 days for the Arctic average. The global BC lifetime in the simulations is within the range of previous estimates, as summarized in Liu and Matsui (2021b). $RE_{BC\_SNOW}$ has large values (>1 W m$^{-2}$) in high mountain areas in the mid-latitudes, Siberia, and coastal areas of southern Greenland (Fig. 6e) and the global and Arctic averages are estimated to be 0.047 W m$^{-2}$ and 0.19 W m$^{-2}$.

Some previous studies have estimated the burden and direct radiative forcing (preindustrial to present-day) of BC for north of 60°N. In this study, $M_{BC\_COL}$ is estimated to be 0.0043 Tg (>60°N), which is slightly lower than the range of previous estimates (e.g., 0.0054−0.0091 Tg in Mahmood et al., 2016). The direct radiative forcing of anthropogenic BC at TOA is 0.17 W m$^{-2}$ (>60°N), which is within the range of the Aerosol Comparisons between Observations and Models (AeroCom) modelled estimates of 0.03−0.37 W m$^{-2}$ (median 0.19 W m$^{-2}$) in Sand et al. (2017). $RE_{BC\_SNOW}$ is 0.21 W m$^{-2}$ (>60°N), which is also consistent with the AeroCom estimate of 0.17 W m$^{-2}$ (0.06−0.28 W m$^{-2}$) reported in Jiao et al. (2014).

## 3.3 Source contributions of Arctic BC

The estimated source contributions to the five variables ($M_{BC\_SRF}$, $M_{BC\_COL}$, $M_{BC\_DEP}$, $RE_{BC\_TOA}$, and $RE_{BC\_SNOW}$) differ greatly (Fig. 7). For $M_{BC\_SRF}$, the contribution from Siberia is dominant (70%), followed by Europe (12%) and Asia (Central Asia + East Asia + Southeast Asia) (8.6%). Anthropogenic sources account for 94% of the total. In contrast, for $M_{BC\_COL}$, the contribution from Asia accounts for 37%, which is larger than the contributions from Siberia (34%) and Europe (13%). The larger contribution from Asia is due to BC transport from high-latitude (nearby) sources being dominant near the surface, whereas the contribution of BC transported over long distances from the mid-latitudes is larger in the middle and upper troposphere in the Arctic (e.g., Stohl, 2006).

The major contribution from Siberia to $M_{BC\_SRF}$ estimated in this study is consistent with some previous studies (e.g., Ikeda et al., 2017). In contrast, other studies have reported a large contribution from Europe and North America to $M_{BC\_SRF}$ (e.g., Wang et al., 2014). This difference is due at least partly to the different definitions of the Siberian region among studies and the different years of emissions used (e.g., 2010 in this study and 2000 in earlier studies). The contribution of Siberia is also strongly dependent on the choice of emission inventories because there is a large uncertainty in BC emissions in Siberia; for example, Huang et al. (2015) estimated the largest contribution to be from gas flaring, whereas Winiger et al. (2017) suggested that domestic and transport sources are more important in Siberia than gas flaring.

BC emitted from Asian regions south of 30°N (Central Asia 1 and 2, East Asia 1, and Southeast Asia) accounts for 1.1% of the total $M_{BC\_SRF}$ and 10% of the total $M_{BC\_COL}$ in the Arctic. Previous modelling studies have reported that BC emitted from low latitudes in Asia (e.g., Southeast Asia) can be transported to the Arctic (e.g., Koch and Hansen, 2005; Zhao et al., 2021). In the model simulations in this study, however, the contribution of BC emitted from low latitudes (south of 30°N) to the Arctic region is small (Table 2).

The contribution of mid-latitude (high-latitude) sources to $M_{BC\_DEP}$ is larger (smaller) than that to $M_{BC\_SRF}$ and smaller (larger) than that to $M_{BC\_COL}$. The largest contribution to $M_{BC\_DEP}$ is from Siberia (53%), followed by North America, Europe, East Asia, and Central Asia. Because BC deposition is caused mainly by cloud and precipitation processes, the source contribution to $M_{BC\_DEP}$ depends on the source contribution to atmospheric BC at the altitude where clouds exist (e.g., mainly 2−4 km at Barrow; Mori et al., 2020). $M_{BC\_SRF}$ and $M_{BC\_COL}$ show seasonal variations with maxima in winter and spring (black lines in Figs. 8a and b), whereas $M_{BC\_DEP}$ shows the seasonal variation with a maximum in summer (black line in Fig. 8c). Because the contribution of BC from biomass burning sources is large in summer (Fig. 8c), their annual mean contribution to $M_{BC\_DEP}$ (34%; mainly from Siberia and North America) is larger than that to $M_{BC\_SRF}$ and $M_{BC\_COL}$ (5.9% and 17%, respectively). This result is consistent with recent isotope-based observations showing that the contribution of biomass burning sources to snow BC is larger than their contribution to atmospheric BC (Rodriguez et al., 2020).

The contribution of Asia (East Asia + Central Asia + Southeast Asia) to $RE_{BC\_TOA}$ is 43%, which is larger than its contribution to $M_{BC\_SRF}$ (8.6%) and $M_{BC\_COL}$ (37%). In contrast, the contributions of anthropogenic BC from Siberia and Europe to $RE_{BC\_TOA}$ are 14% and 8.2%, respectively, which are smaller than their contributions to $M_{BC\_COL}$ (26% and 13%, respectively). These results are obtained because the radiative effect per unit $M_{BC\_COL}$ (the radiative effect normalized by $M_{BC\_COL}$ ($NRE_{COL}$)) for anthropogenic BC from Asia is larger than that for anthropogenic BC from Siberia and Europe, as discussed in Sect. 3.4. The contribution of biomass burning sources to $RE_{BC\_TOA}$ (29%) is larger than that to $M_{BC\_COL}$ (17%), because $M_{BC\_COL}$ is higher in winter and early spring when anthropogenic sources dominate (Fig. 8b) whereas $RE_{BC\_TOA}$ is largest in late spring and summer when the contribution of biomass burning sources is large (Fig. 8d). Annual-mean source contributions are therefore significantly different between $RE_{BC\_TOA}$ and $M_{BC\_COL}$ (Fig. 7).

The source contributions to $RE_{BC\_SNOW}$ are generally similar to those to $M_{BC\_DEP}$. The contribution from Siberia is largest (41%), followed by North America (>50°N) (24%) and Asia (19%) (Fig. 7). The contributions of these sources are large in both online and offline calculations (Sect. 2, Fig. S2). Because $RE_{BC\_SNOW}$ in the Arctic is largest in late spring and summer in this study (Fig. 8e), the contribution of biomass burning sources to $RE_{BC\_SNOW}$ (35%) is larger than that to atmospheric concentrations (5.9% and 17% for $M_{BC\_SRF}$ and $M_{BC\_COL}$, respectively).

The source contributions to the five variables differ significantly not only on an annual average basis but also on a monthly basis (Fig. 8). The contribution of anthropogenic BC from Siberia to $M_{BC\_SRF}$ reaches 75% in winter (December−February) (Fig. 8a). The contribution of Asia (East Asia + Central Asia + Southeast Asia) to $M_{BC\_SRF}$ is less than 15% throughout the year, whereas its contributions to $M_{BC\_COL}$ and $RE_{BC\_TOA}$ are large in winter and spring: 52% to $M_{BC\_COL}$ and 63% to $RE_{BC\_TOA}$ in March (Figs 8b and d). The contributions of biomass burning sources to the five variables are largest in summer, 12−34% from Siberia and 19−41% from North America (>50°N) (June−August average). The large contribution of biomass burning sources to $M_{BC\_SRF}$ and $M_{BC\_COL}$ during summer is consistent with previous studies (e.g., Winiger et al., 2019).

Figure 9 shows the spatial distributions of the source regions with the largest contributions to BC among nine source regions: Europe, Siberia, Greenland, North America (>50°N), North America (<50°N), Central Asia, East Asia, Southeast Asia, and Others. The contribution of each emission source is largest near the source. Sources making the largest contributions to Arctic BC differ significantly among the five variables. For $M_{BC\_SRF}$, Siberia's contribution is the largest over 77% of the total Arctic area (Fig. 9a), followed by Europe (14%) and North America (>50°N) (8.9%). For $M_{BC\_DEP}$, Siberia's contribution is the largest over 60% of the total Arctic area (Fig. 9c), followed by North America (>50°N) (30%), and Europe (10%). There is no area of the Arctic where the contribution of East Asia is the largest for $M_{BC\_SRF}$ and $M_{BC\_DEP}$.

Unlike $M_{BC\_SRF}$ and $M_{BC\_DEP}$, for $M_{BC\_COL}$, the Arctic area where the contribution of East Asia is the largest extends over the North American side of the Arctic (38% of the Arctic area) (Fig. 9b), and the area where Siberia's contribution is the largest extends over the Siberian side of the Arctic (53% of the Arctic area). For $RE_{BC\_TOA}$, the contribution from East Asia (Siberia) is the largest over 56% (39%) of the Arctic region (Fig. 9d). For $RE_{BC\_SNOW}$, which is limited to land areas, North America's (>50°N) contribution is the largest over 53% of the Arctic area (over the North American side of the Arctic), and the

contributions of Siberia, Europe, and East Asia are the largest over 40%, 4.5%, and 3.2%, respectively, of the Arctic area (over the Siberian side of the Arctic) (Fig. 9e).

The source contributions of BC show year-to-year variability, mainly in response to interannual variations in BC emissions at mid- and high latitudes (Fig. 10). For the years 2012, 2015, and 2016, BC emissions from biomass burning sources north of 50°N are about twice those for the other years, and the contributions from biomass burning sources to $M_{BC\_COL}$ and $RE_{BC\_TOA}$ are larger in the Arctic (Figs. 10b and 10d). The contributions from biomass burning sources in Siberia and North America (>50°N) to $M_{BC\_DEP}$, $RE_{BC\_TOA}$, and $RE_{BC\_SNOW}$ vary between years by a factor of 3.4 to 6.4 (by up to about 20%), with large

interannual variability (Fig. 10, Table S1). Compared with those of biomass burning BC, the source contributions of anthropogenic BC show smaller interannual variability: source contributions generally vary within a factor of 2 (within 10%). Our anthropogenic BC emissions north of 50°N decrease by about 10% from 2009 to 2015 (Fig. S6a). In addition, the atmospheric lifetime of anthropogenic BC north of 50°N is longest in 2009 (Fig. S6b). For these reasons, the source contribution of anthropogenic BC is largest in 2009 and tends to decrease in subsequent years (Fig. 10). Overall, the source

contributions to the five BC variables show interannual variation to some extent, but the qualitative source characteristics (e.g., which sources make large contributions) do not change significantly during the simulation periods.

    In summary, the results shown in this section demonstrate that the source contributions (Figs. 7 and 8) and the spatial distributions of the areas making the largest contributions (Fig. 9) to Arctic BC differ substantially among $M_{BC\_SRF}$, $M_{BC\_COL}$, $M_{BC\_DEP}$, $RE_{BC\_TOA}$, and $RE_{BC\_SNOW}$ for all years simulated.

**3.4 Different radiative effect efficiency among sources**

CAM-ATRAS uses 47 bins for each of the 26 emission sources to calculate the particle size and mixing state of BC for each source (Sect. 2). Using this information, in this section, we estimate microphysical properties, absorption aerosol optical depth (AAOD), and radiative effects of BC for all emission sources and investigate their differences. $RE_{BC\_TOA}$ can be decomposed into three components by Eq. 2 (Matsui et al., 2018b):

$RE_{BC\_TOA} = M_{BC\_COL} \times \frac{AAOD_{BC}}{M_{BC\_COL}} \times \frac{RE_{BC\_TOA}}{AAOD_{BC}} = M_{BC\_COL} \times MAC_{BC} \times NRE_{AAOD}$ ,          (2)

where $AAOD_{BC}$ is the AAOD of BC at the wavelength of 550 nm. $MAC_{BC}$ is defined as the ratio of $AAOD_{BC}$ to $M_{BC\_COL}$. The BC radiative effect normalized by $AAOD_{BC}$ ($NRE_{AAOD}$) is defined as the ratio of $RE_{BC\_TOA}$ to $AAOD_{BC}$. The BC radiative effect normalized by $M_{BC\_COL}$ ($NRE_{COL}$; $RE_{BC\_TOA}$ / $M_{BC\_COL}$ or $MAC_{BC} \times NRE_{AAOD}$) is also used. The global-mean $NRE_{AAOD}$ and $NRE_{COL}$ in this study are 151 W m$^{-2}$ and 1317 W g$^{-1}$, respectively; these values are consistent with the median values of

130 W m$^{-2}$ (84−216 W m$^{-2}$) and 1322 W g$^{-1}$ (612−2661 W g$^{-1}$) in the AeroCom Phase II models (Myhre et al., 2013).

    Figure 11 shows annual-mean $MAC_{BC}$, $NRE_{AAOD}$, and $NRE_{COL}$ values in the Arctic for eight major BC sources (six anthropogenic and two biomass burning sources). Anthropogenic BC from Europe, Siberia, and North America (>50°N) (6.7−7.3 m$^2$ g$^{-1}$) has lower $MAC_{BC}$ than ALL BC (8.4 m$^2$ g$^{-1}$) (Fig. 11a, Table 2), whereas anthropogenic BC from Asia (Central Asia and East Asia) and biomass burning BC from Siberia and North America (>50°N) have higher $MAC_{BC}$ values

(8.5−9.4 m$^2$ g$^{-1}$). These differences in $MAC_{BC}$ are because the mixing state of BC from each emission source differs. Compared with anthropogenic BC from Siberia, Europe, and North America (>50°N), anthropogenic BC particles from Asia have a higher fraction of thickly coated BC particles (which have higher $MAC_{BC}$) and a lower fraction of thinly coated BC particles (which have lower $MAC_{BC}$) (Fig. 12a). The higher fraction of thickly coated BC from Asia might be explained by fast aging processes near their sources, where the concentrations of condensable gases emitted with BC are high, and by the higher

fraction of anthropogenic BC from Asia in the upper troposphere in the Arctic (Fig. S7) and its longer lifetime in the Arctic (24−30 days) (Fig. 12b). Biomass burning BC from Siberia and North America (>50°N) also has a higher fraction of thickly coated BC particles and a lower fraction of thinly coated BC particles than anthropogenic BC from Siberia, Europe, and North America (>50°N) (Fig. 12a), possibly because BC aging processes are faster in summer, when the contribution of biomass burning sources is larger, than in winter. The fraction of thickly coated BC was observed to be high in the Arctic in recent

aircraft measurements by Ohata et al. (2021b), consistent with our model simulations, although it is difficult to observe the dependence of BC mixing states on emission sources. Our simulation results indicate the importance of understanding the differences in BC mixing states among sources and the mechanisms that control them in evaluating the source contribution of BC to $RE_{BC\_TOA}$ in the Arctic.

Similar to $MAC_{BC}$, $NRE_{AAOD}$ also differs substantially among emission sources. The $NRE_{AAOD}$ value of anthropogenic BC from Europe and Siberia (210−245 W m$^{-2}$, Arctic average) is 29−39% lower than that of ALL BC (345 W m$^{-2}$), whereas that of anthropogenic BC from Asia and biomass burning BC from Siberia and North America (>50°N) is 2.1−65% higher (353−568 W m$^{-2}$). $NRE_{AAOD}$ depends on altitude, solar radiation, and surface albedo where BC exists. A higher fraction of BC at high altitudes and where the surface albedo is higher leads to a higher $NRE_{AAOD}$ value (e.g., Samset and Myhre, 2015). Solar radiation in the Arctic is highest during the summer. Here, BC-concentration-weighted mean height ($Height_{BC}$), mean solar radiation (downward radiation flux at TOA) ($Flux_{BC}$), and mean surface albedo ($Albedo_{BC}$) in the Arctic are defined for each emission source as follows:

$$Height_{BC,s} = \frac{\sum_{i,k,m,s} M_{BC,i,k,m,s} \times Height_{i,k,m}}{\sum_{i,k,m,s} M_{BC,i,k,m,s}}, \tag{3}$$

$$Flux_{BC,s} = \frac{\sum_{i,m,s} M_{BC\_COL,i,m,s} \times Flux_{i,m}}{\sum_{i,m,s} M_{BC\_COL,i,m,s}}, \tag{4}$$

$$Albedo_{BC,s} = \frac{\sum_{i,m,s} M_{BC\_COL,i,m,s} \times Albedo_{i,m}}{\sum_{i,m,s} M_{BC\_COL,i,m,s}}, \tag{5}$$

where $M_{BC,i,k,m,s}$ denotes the BC mass concentration in horizontal grid $i$, vertical grid $k$, month $m$, and emission source $s$; $M_{BC\_COL,i,m,s}$ denotes $M_{BC\_COL}$ in horizontal grid $i$, month $m$, and emission source $s$; and $Height_{i,k,m}$, $Flux_{i,m}$, and $Albedo_{i,m}$ are height (above sea level), solar radiation flux, and surface albedo, respectively, in each grid and month. These equations are calculated for grids in the Arctic (>70°N) to derive the mean height ($Height_{BC,s}$), mean solar radiation flux ($Flux_{BC,s}$), and mean surface albedo ($Albedo_{BC,s}$), weighted by the BC mass concentration from each emission source $s$. The $Height_{BC}$ values of anthropogenic BC from Asia (Central Asia + East Asia) (>3500 m in the Arctic) are higher than those of anthropogenic BC from Europe and Siberia (<2000 m) (Fig. 13a). In addition, the $Flux_{BC}$ values of anthropogenic BC from Asia (157−185 W m$^{-2}$ in the Arctic) are 20−40% higher than those of anthropogenic BC from Europe and Siberia (122−140 W m$^{-2}$) (Fig. 13b). For these reasons, the $NRE_{AAOD}$ values of anthropogenic BC from Asia are higher than those of anthropogenic BC from Europe and Siberia (Fig. 11b). The $Flux_{BC}$ values of biomass burning BC from Siberia and North America (>50°N) are 90−130% (1.9−2.3 times) higher than those of ALL BC in the Arctic (Fig. 13b), owing to the larger amounts of biomass burning BC in summer when solar radiation flux is the highest in the Arctic. These higher $Flux_{BC}$ values of biomass burning BC are the main reason why biomass burning BC has a higher $NRE_{AAOD}$ than anthropogenic BC (Fig. 11b).

$NRE_{COL}$ (the product of $MAC_{BC}$ and $NRE_{AAOD}$) of ALL BC in the Arctic in this study is 2888 W g$^{-1}$, which is lower than the values of around 3000−5000 W g$^{-1}$ in the AeroCom models (Samset et al., 2013). This lower value is likely because the fraction of Arctic BC existing at lower altitudes is higher in this study (70% above 500 hPa (below ~5 km)) than in AeroCom models (~40% below 5 km). $NRE_{COL}$ values of anthropogenic BC from Europe and Siberia are lower (1471−1781 W g$^{-1}$), and those of anthropogenic BC from Asia (3064−3351 W g$^{-1}$) and biomass burning BC from Siberia and North America (>50°N) (4148−5326 W g$^{-1}$) are higher (Fig. 11c). $NRE_{COL}$ of anthropogenic BC from Central Asia is 130% (2.3 times) larger than that of anthropogenic BC from Siberia. $NRE_{COL}$ of biomass burning BC from Siberia and North America (>50°N) is 180% (2.8 times) and 260% (3.6 times) higher, respectively, than that of anthropogenic BC from Siberia. Thus, $NRE_{COL}$ ($RE_{BC\_TOA}$ per unit BC mass) in the Arctic differs by a factor of up to about 4 among the emission sources because mixing states, heights, and seasonal variations (solar radiation) are different.

## 4 Summary

In this study, we estimate the source contributions of Arctic BC to five BC variables, $M_{BC\_SRF}$, $M_{BC\_COL}$, $M_{BC\_DEP}$, $RE_{BC\_TOA}$, and $RE_{BC\_SNOW}$, and show that the source contributions differ significantly among them. $M_{BC\_SRF}$ is dominated by Siberian sources (70%), whereas the contribution from Siberia (34%) to $M_{BC\_COL}$ is smaller than that from Asia (37%). These differences can be attributed to the fact that BC transport from high-latitude emission sources is dominant in the lower troposphere in the Arctic, whereas long-range BC transport from mid-latitudes is more important in the middle and upper troposphere in the Arctic. The contributions from Siberia and Asia to $M_{BC\_DEP}$ are 53% and 15%, respectively. The contributions of biomass burning sources to $M_{BC\_SRF}$, $M_{BC\_COL}$, and $M_{BC\_DEP}$ are larger during summer months. Because $M_{BC\_DEP}$ is highest in summer, the contribution from biomass burning sources to $M_{BC\_DEP}$ is larger (20% from Siberia and 12% from North America (>50°N)) than that to $M_{BC\_SRF}$ and $M_{BC\_COL}$. The contribution from Asia (Siberia) to $RE_{BC\_TOA}$ is 43% (26%), which is larger (smaller) than its contribution to $M_{BC\_COL}$. The contribution from biomass burning to $RE_{BC\_TOA}$ is also large (29%). The contribution from Siberia to $RE_{BC\_SNOW}$ is 41%, which is larger than its contribution to $RE_{BC\_TOA}$. $RE_{BC\_TOA}$ (from all sources) is 0.40 W m$^{-2}$ globally and 0.31 W m$^{-2}$ in the Arctic. $RE_{BC\_SNOW}$ is 0.047 W m$^{-2}$ globally and 0.19 W m$^{-2}$ in the Arctic.

We also show that the radiative effect efficiency of BC ($NRE_{COL}$; $RE_{BC\_TOA}$ / $M_{BC\_COL}$) in the Arctic from each emission source differs by a factor of up to about 4 (1471−5326 W g$^{-1}$). Anthropogenic BC from Asia and biomass burning BC from Siberia and North America (>50°N) have a higher fraction of thickly coated BC particles and higher $MAC_{BC}$ ($AAOD_{BC}$ / $M_{BC\_COL}$) at the wavelength of 550 nm (8.5−9.4 m$^2$ g$^{-1}$). In contrast, anthropogenic BC from Europe, Siberia, and North America (>50°N) has a higher fraction of thinly coated BC particles and lower $MAC_{BC}$ (6.7−7.3 m$^2$ g$^{-1}$). $MAC_{BC}$ in the Arctic differs by up to 41% among emission sources. $NRE_{AAOD}$ ($RE_{BC\_TOA}$ / $AAOD_{BC}$) also differs significantly among emission sources because the altitude of BC and incident solar radiation flux (i.e., seasonal variations) are different. $NRE_{AAOD}$ of anthropogenic BC from Asia and biomass burning BC from Siberia and North America (>50°N) is up to 170% (2.7 times) greater than that of anthropogenic BC from Siberia and North America (>50°N). As a result, $NRE_{COL}$ (product of $MAC_{BC}$ and $NRE_{AAOD}$) in the Arctic differs by up to 3.6 times among emission sources.

The results of this study demonstrate that source contributions to BC in the Arctic differ substantially depending on BC variables. The contribution of Asia to $RE_{BC\_TOA}$ is the largest, whereas Siberia makes the largest contribution to $RE_{BC\_SNOW}$. The source contributions to $RE_{BC\_TOA}$ and $RE_{BC\_SNOW}$ are quite different from the source contributions to $M_{BC\_SRF}$, $M_{BC\_COL}$, and $M_{BC\_DEP}$. The results also demonstrate the importance of accurately estimating the differences in microphysical properties (e.g., mixing state), altitude, seasonal variations, and the resulting radiative effect efficiency of BC ($NRE_{COL}$) from different emission sources when estimating the source contributions of BC radiative effects.

**Acknowledgements**

This work was supported by the Ministry of Education, Culture, Sports, Science and Technology of Japan and the Japan Society for the Promotion of Science (MEXT/JSPS) KAKENHI Grant Numbers JP18H03363, JP19H04253, JP19H05699, JP19KK0265, JP20H00196, JP20H00638, and JP22H03722, by the MEXT Arctic Challenge for Sustainability phase II (ArCS-II; JPMXD1420318865) project, and by the Environment Research and Technology Development (JPMEERF20202003) of the Environmental Restoration and Conservation Agency of Japan. This work was also supported by Nagoya University Research Fund and a grant for the Global Environmental Research Coordination System from the Ministry of the Environment, Japan (MLIT1753). We thank the U.S. National Oceanic and Atmospheric Administration (NOAA) Black Carbon Group for providing us their BC data from aircraft measurements.

**Author contribution**

H.M. conceived and designed the research, performed model simulations and data analysis, and wrote the manuscript. T.M., S.O., N.M., N.O., K.G.-A., M.K., and Y.K. made BC observations for surface atmosphere at Barrow, Ny-Ålesund, Alert, and
455 Pallas and for snow in Finland, Alaska, Siberia, and Greenland and at Ny-Ålesund. N.M., S.O., M.K., and Y.K. made BC observations during the ARCTAS and PAMARCMiP2018 aircraft campaigns. All authors interpreted data, discussed their implications, and contributed to the manuscript.

**Code/Data availability**

Data used in this study are available upon request from the corresponding author (H.M.).

**Competing interests**

The authors declare no conflict of interests.

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

**Figures**

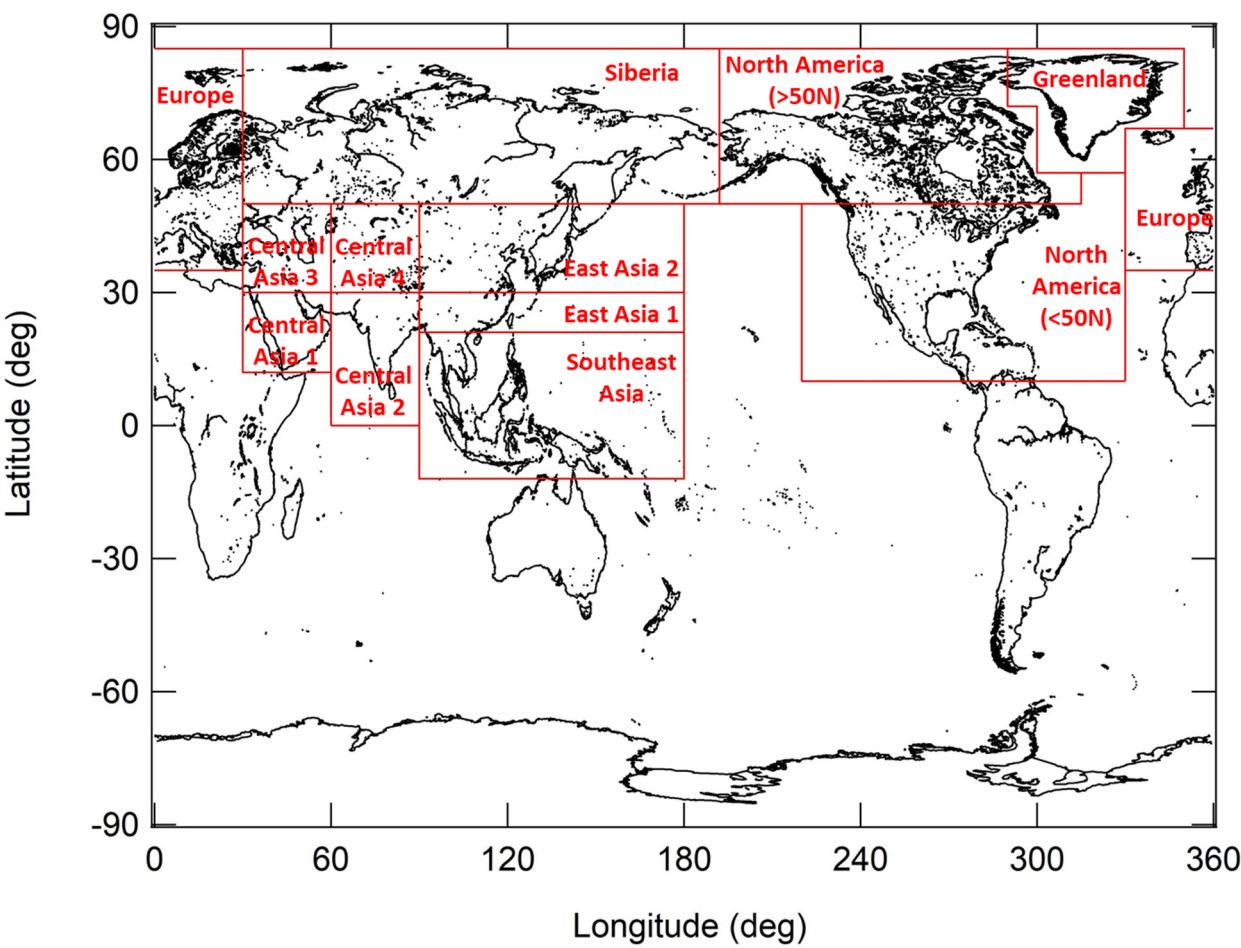

**Figure 1: The definition of source regions used in this study. The 13 source regions are Europe (EUR), Siberia (SIB), Greenland (GL), North America north of 50°N (NAM (>50°N)), North America south of 50°N (NAM (<50°N)), Central Asia (CAS) 1−4, East Asia (EAS) 1−2, Southeast Asia (SAS), and Others. Anthropogenic and biomass burning BC from each source region are tracked by tag tracers in global aerosol model simulations using CAM-ATRAS (Sect. 2).**


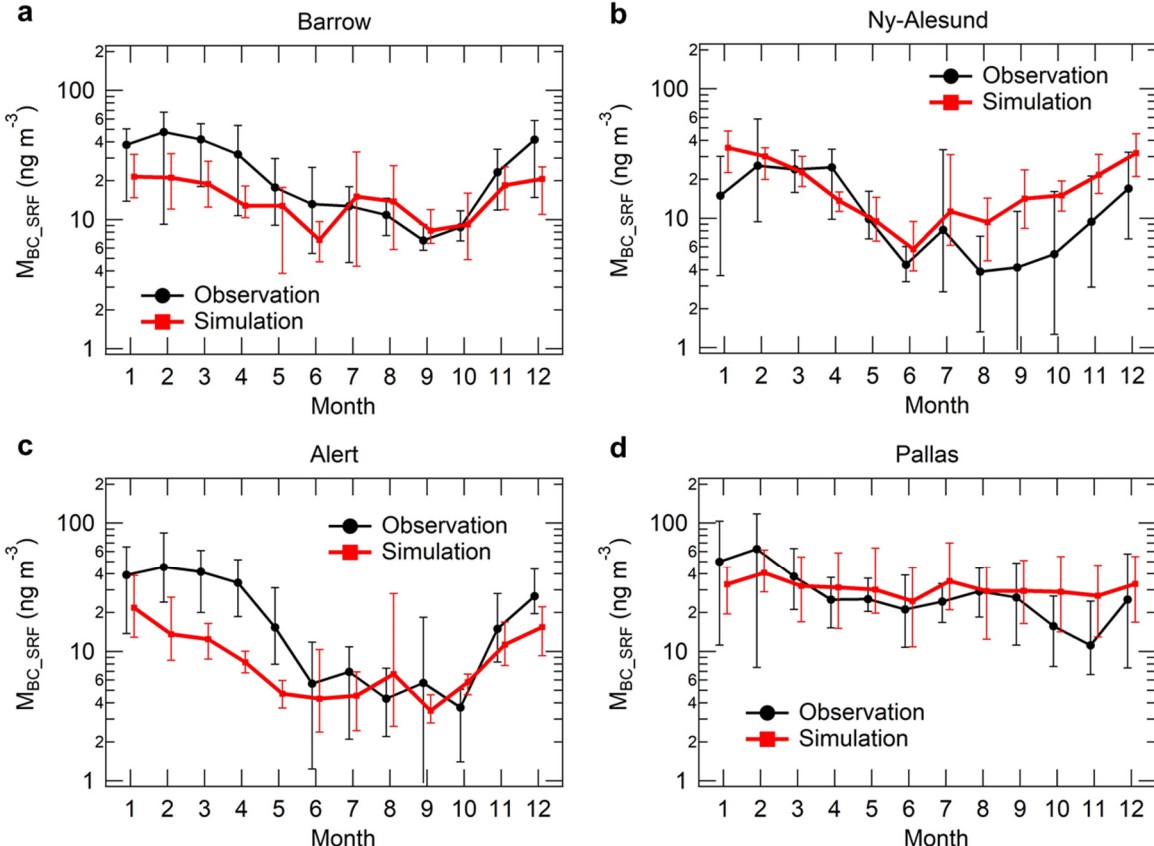


**Figure 2: Comparisons between observations (black) and model simulations (red) for surface BC mass concentrations ($M_{BC\_SRF}$) at (a) Barrow, (b) Ny-Ålesund, (c) Alert, and (d) Pallas. Model simulations in 2009−2015 were compared with observations in 2009−2015. The error bars show the interannual variability (maximum−minimum ranges) of $M_{BC\_SRF}$. $M_{BC\_SRF}$ is shown at standard temperature and pressure in both observations and model simulations.**


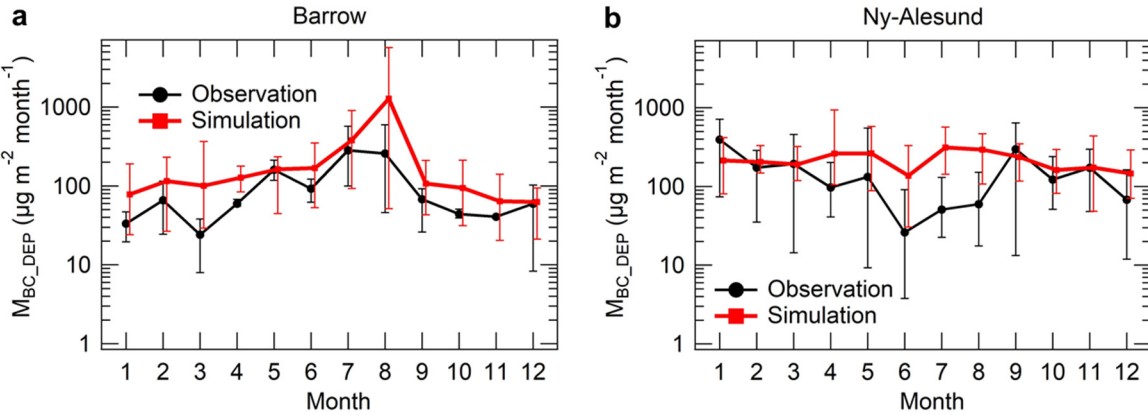

**Figure 3: Comparisons between observations (black) and model simulations (red) for BC deposition flux ($M_{BC\_DEP}$) at (a) Barrow and (b) Ny-Ålesund. Model simulations in 2009−2015 were compared with observations in 2013−2017 because $M_{BC\_DEP}$ observation data are available during 2013−2017. The error bars show the interannual variability (maximum−minimum ranges) of $M_{BC\_DEP}$.**

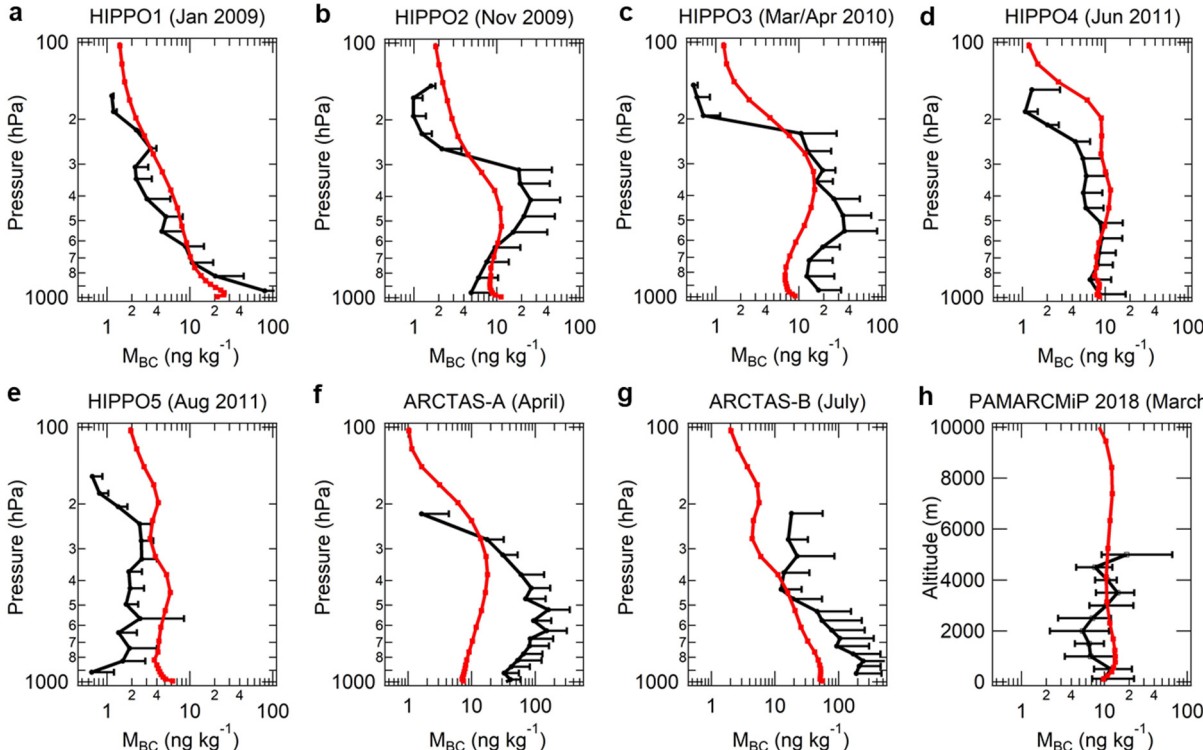

**Figure 4: Comparisons between observations (black) and model simulations (red) for BC mass concentration ($M_{BC}$) vertical profiles at high latitudes in the Northern Hemisphere during (a−e) the HIPPO campaigns ((a) January 2009, (b) November 2009, (c) March−April 2010, (d) June 2011, and (e) August 2011), (f−g) the ARCTAS campaigns ((f) April 2008 and (g) July 2008), and the PAMARCMiP campaign in March−April 2018. For the HIPPO campaigns, simulated $M_{BC}$ concentrations are averaged over the region of 60−80°N and 140−170°W for the observation year and month. For the other campaigns, simulated $M_{BC}$ concentrations are averaged for 7 years (2009−2015) over the regions of 60−80°N and 70−165°W in April for ARCTAS-A, 45−87°N and 40−135°W in July for ARCTAS-B, and 78−85°N and 24°W−20°E in March for PAMARCMiP. For the observed $M_{BC}$, the means and standard deviations are shown against atmospheric pressure for the HIPPO and ARCTAS campaigns, and the medians and 25th−75th percentiles are shown against altitude for the PAMARCMiP campaign.**

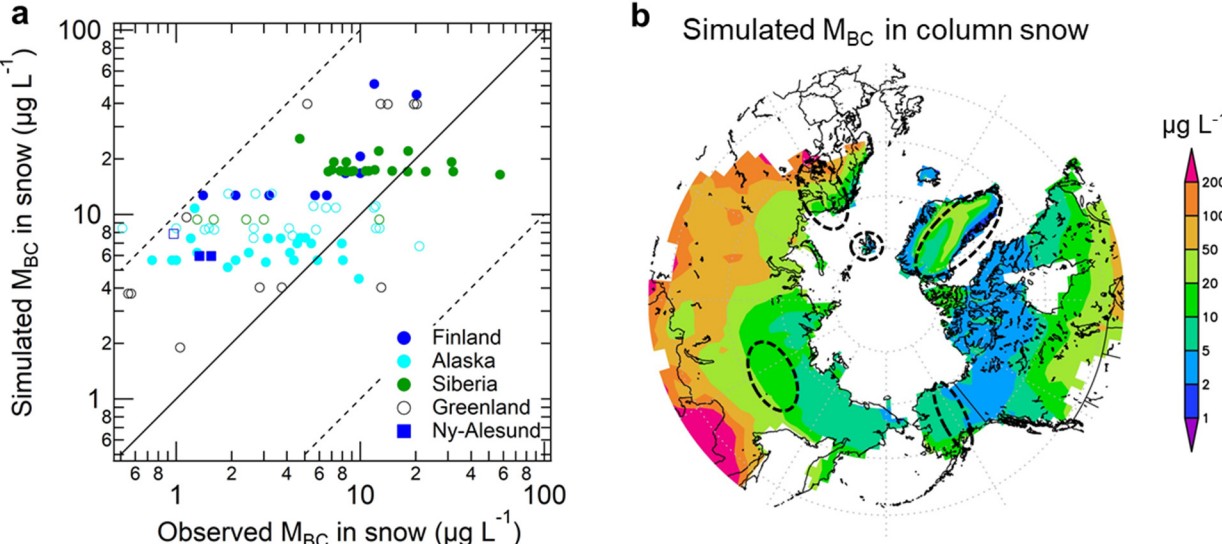

**Figure 5: (a) Scatter plot of observed and simulated BC mass concentrations ($M_{BC}$) in snow in Finland (blue, circles), Alaska (light blue), Siberia (green), Greenland (black), and Ny-Ålesund (blue, squares). Observed data are taken from Mori et al. (2019). Simulation results (monthly averages) are shown for individual sampling points (latitude, longitude) and periods (years, months). Closed and open circles indicate $M_{BC}$ in column snow and surface snow, respectively. The 1:1 line (solid black line) and the 10:1 and 1:10 lines (dashed lines) are also shown. (b) Simulated $M_{BC}$ in column snow at high latitudes in the Northern Hemisphere in March (2009−2015). Dashed circles indicate the approximate area where snow samplings were performed (Mori et al., 2019).**

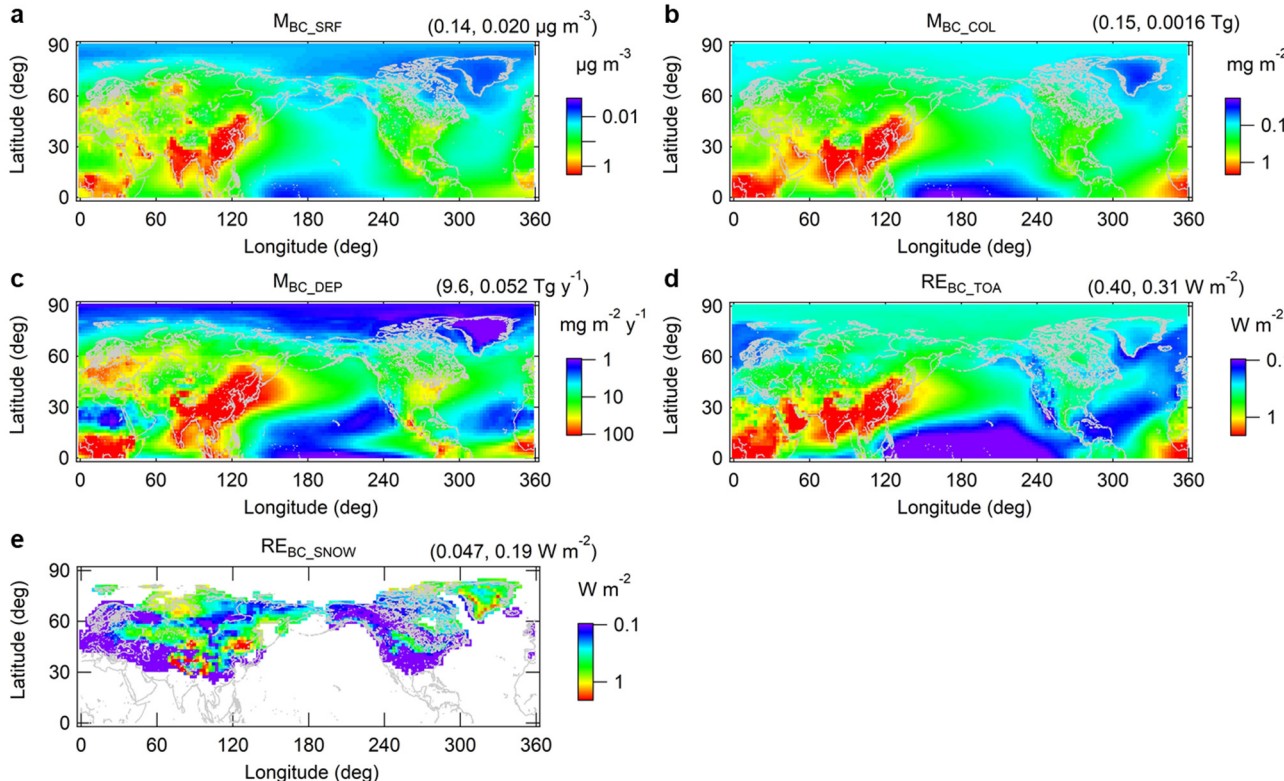

**Figure 6: Spatial distributions of (a) M$_{BC\_SRF}$, (b) M$_{BC\_COL}$, (c) M$_{BC\_DEP}$, (d) RE$_{BC\_TOA}$, and (e) RE$_{BC\_SNOW}$ in the Northern Hemisphere. The values in parentheses are global (left) and Arctic (right) mean values (annual mean). Purple shows areas where values are below** 790 **the minimum shown on the colour bars.**

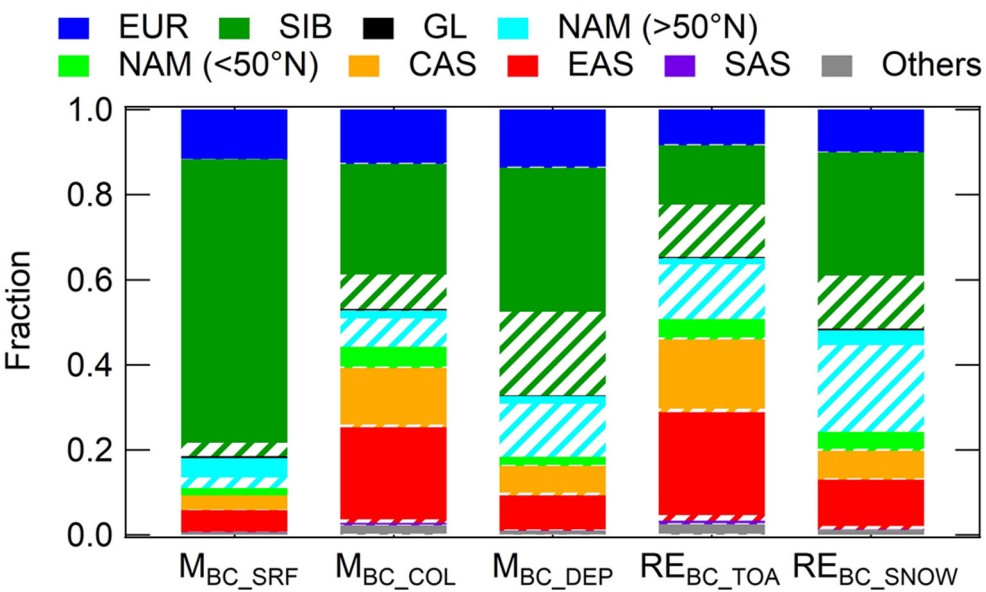

Figure 7: Source contributions to $M_{BC\_SRF}$, $M_{BC\_COL}$, $M_{BC\_DEP}$, $RE_{BC\_TOA}$, and $RE_{BC\_SNOW}$ (from left to right) in the Arctic (annual mean). The filled and shaded areas indicate contributions from anthropogenic and biomass burning sources, respectively. EUR, SIB, GL, NAM, CAS, EAS, and SAS denote Europe, Siberia, Greenland, North America, Central Asia, East Asia, and Southeast Asia, respectively.

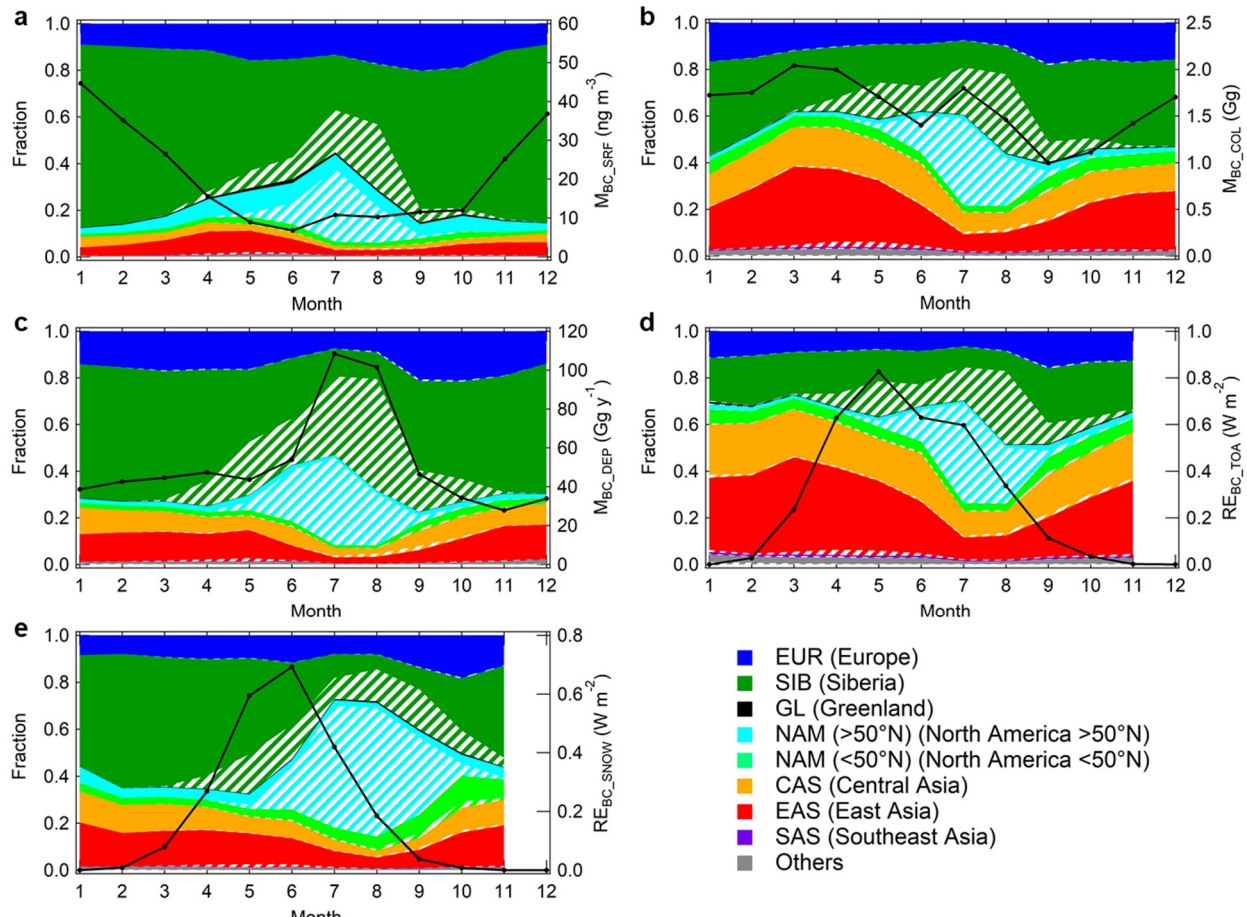

**Figure 8: Monthly variations of source contributions to (a) M$_{BC\_SRF}$, (b) M$_{BC\_COL}$, (c) M$_{BC\_DEP}$, (d) RE$_{BC\_TOA}$, and (e) RE$_{BC\_SNOW}$ in the Arctic. The filled and shaded areas indicate contributions from anthropogenic and biomass burning sources, respectively. The black lines (right axis) show total BC concentrations, deposition flux, or radiative effects from all sources.**

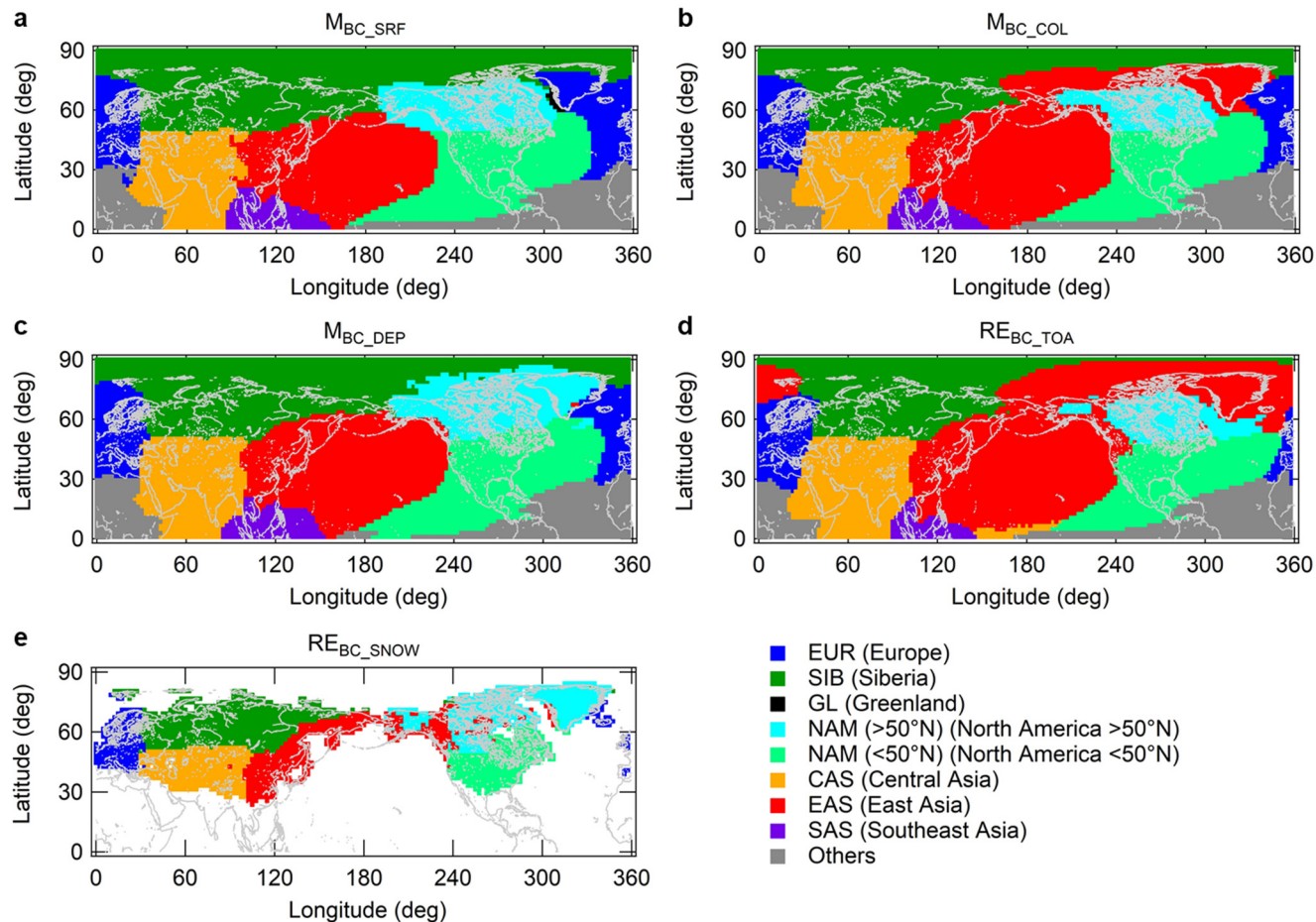

Figure 9: Spatial distributions of emission sources with the largest contribution to (a) $M_{BC\_SRF}$, (b) $M_{BC\_COL}$, (c) $M_{BC\_DEP}$, (d) $RE_{BC\_TOA}$, and (e) $RE_{BC\_SNOW}$ among the nine source regions.

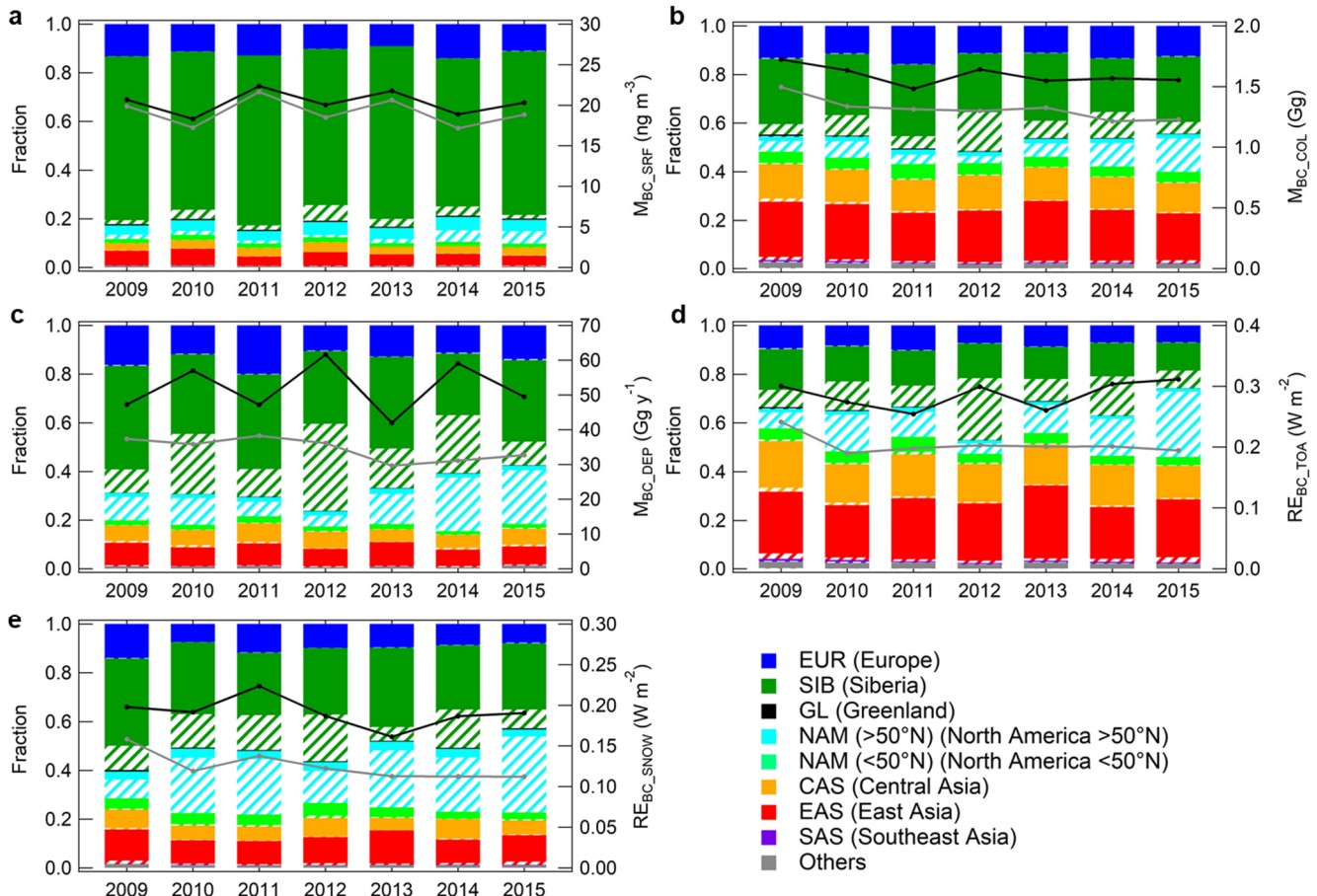

**Figure 10: Year-to-year variations of annual-mean source contributions to (a) M**BC_SRF**, (b) M**BC_COL**, (c) M**BC_DEP**, (d) RE**BC_TOA**, and (e) RE**BC_SNOW** in the Arctic for the years from 2009 to 2015 (left axis). The filled and shaded areas indicate contributions from anthropogenic and biomass burning sources, respectively. The black and grey lines show BC concentrations, deposition flux, or radiative effects from all (anthropogenic + biomass burning) sources and from anthropogenic sources, respectively (right axis).**

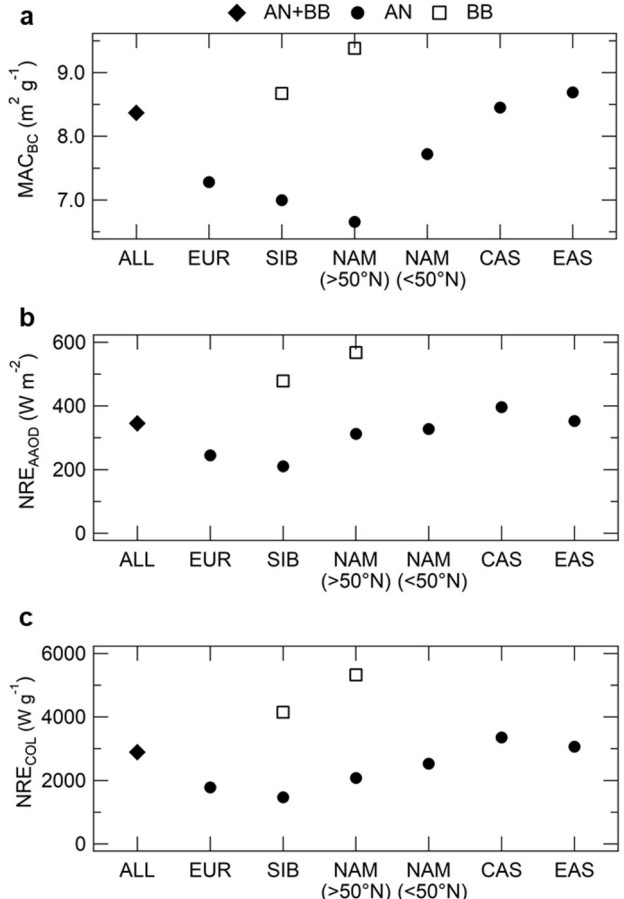

**Figure 11: Optical properties and radiative effect efficiencies in the Arctic for total BC (ALL; from all sources) and BC from the eight major sources (six anthropogenic (AN) sources and two biomass burning (BB) sources): (a) Mass absorption cross section of BC ($MAC_{BC}$), (b) BC radiative effect normalized by absorption aerosol optical depth (AAOD) of BC ($NRE_{AAOD}$; $RE_{BC\_TOA}$ / $AAOD_{BC}$) and (c) BC radiative effect normalized by $M_{BC\_COL}$ ($NRE_{COL}$; $RE_{BC\_TOA}$ / $M_{BC\_COL}$).**

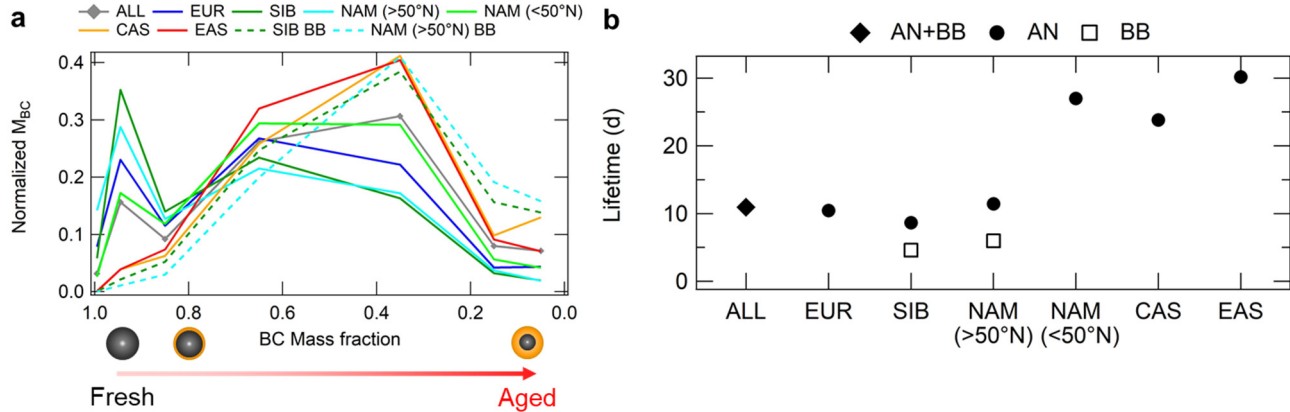

**Figure 12: (a)** BC mixing state distributions for total BC mass (ALL) and for BC mass from the eight major sources (six anthropogenic (AN) sources and two biomass burning (BB) sources). BC particles are gradually shifted from the left (fresh BC with a lower $MAC_{BC}$) to the right (aged BC with a higher $MAC_{BC}$) by aging processes in the atmosphere. **(b)** Lifetimes in the Arctic for total BC (ALL) and for BC from eight major sources. Lifetimes were defined by the ratio of BC deposition flux to atmospheric BC loading in the Arctic.

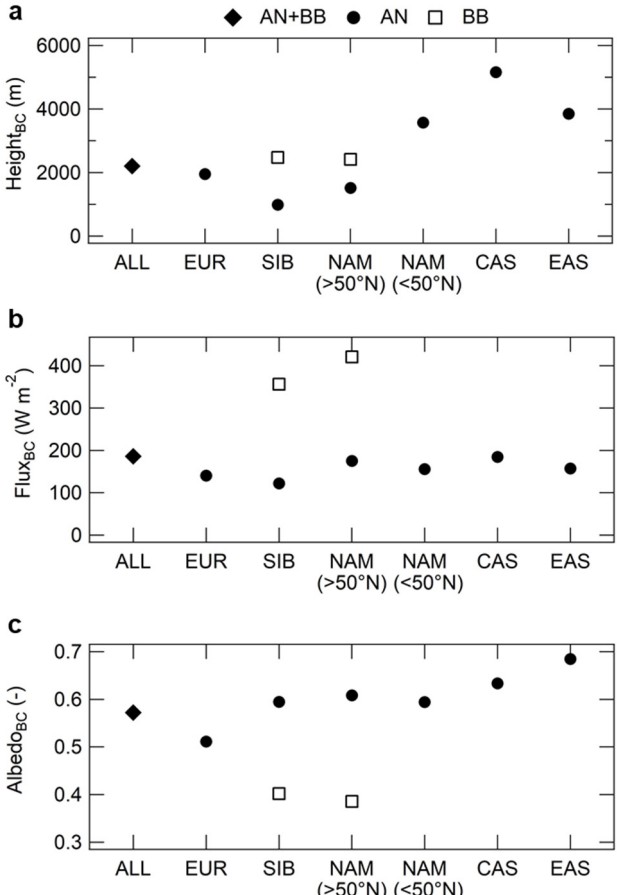

**Figure 13: (a) Mean height (above sea level) (Height$_{BC}$), (b) mean solar radiation flux at the TOA (Flux$_{BC}$), and (c) mean surface albedo (Albedo$_{BC}$) weighted by BC concentrations in the Arctic (Eqs. 3−5) for total BC (ALL) and BC from eight major sources (six anthropogenic (AN) sources and two biomass burning (BB) sources).**

**Tables**

**Table 1: Abbreviations for BC used in this study**

| Terminology | Definition |
|---|---|
| $M_{BC}$ | BC mass concentration |
| $M_{BC\_SRF}$ | Near-surface atmospheric BC mass concentration |
| $M_{BC\_COL}$ | Vertically integrated atmospheric BC mass concentration |
| $M_{BC\_DEP}$ | BC deposition flux |
| $RE_{BC\_TOA}$ | BC radiative effect at the top of the atmosphere |
| $RE_{BC\_SNOW}$ | BC radiative effect on the snow surface |
| $AAOD_{BC}$ | Absorption aerosol optical depth of BC at the wavelength of 550 nm |
| $MAC_{BC}$ | Mass absorption cross section of BC ($AAOD_{BC} / M_{BC\_COL}$) |
| $NRE_{AAOD}$ | BC radiative effect normalized by $AAOD_{BC}$ ($RE_{BC\_TOA} / AAOD_{BC}$) |
| $NRE_{COL}$ | BC radiative effect normalized by $M_{BC\_COL}$ ($RE_{BC\_TOA} / M_{BC\_COL}$) |
| $Height_{BC}$ | BC-concentration weighted mean height above sea level |
| $Flux_{BC}$ | BC-concentration weighted mean downward solar radiation flux at the top of atmosphere |
| $Albedo_{BC}$ | BC-concentration weighted mean surface albedo |

**Table 2. Simulated BC statistics in the Arctic (>70°N)**

| Region [a] | Source [b] | $M_{BC\ SRF}$ | $M_{BC\ COL}$ | $M_{BC\ DEP}$ | $RE_{BC\ TOA}$ | $RE_{BC\ SNOW}$ | $MAC_{BC}$ [c] | $NRE_{AAOD}$ [c] | $NRE_{COL}$ [c] |
|---|---|---|---|---|---|---|---|---|---|
| | | ng m$^{-3}$ | Gg | Gg y$^{-1}$ | W m$^{-2}$ | W m$^{-2}$ | m$^2$ g$^{-1}$ | W m$^{-2}$ | W g$^{-1}$ |
| ALL | AN | 19 | 1.3 | 34 | 0.21 | 0.12 | 8.2 | 307 | 2524 |
| | BB | 1.2 | 0.26 | 18 | 0.081 | 0.066 | 9.0 | 517 | 4664 |
| | | % | % | % | % | % | m$^2$ g$^{-1}$ | W m$^{-2}$ | W g$^{-1}$ |
| EUR | AN | 12 | 13 | 13 | 8.2 | 9.9 | 7.3 | 245 | 1781 |
| | BB | 0.061 | 0.21 | 0.28 | 0.21 | 0.17 | --- | --- | --- |
| SIB | AN | 67 | 26 | 34 | 14 | 29 | 7.0 | 210 | 1471 |
| | BB | 3.1 | 8.1 | 20 | 12 | 12 | 8.7 | 478 | 4148 |
| GL | AN | 0.49 | 0.32 | 0.14 | 0.23 | 0.43 | --- | --- | --- |
| | BB | <0.001 | 0.067 | <0.001 | 0.080 | 0.0014 | --- | --- | --- |
| NAM (>50°N) | AN | 4.6 | 1.9 | 1.8 | 1.4 | 3.4 | 6.7 | 312 | 2079 |
| | BB | 2.4 | 6.6 | 12 | 13 | 20 | 9.4 | 568 | 5326 |
| NAM (<50°N) | AN | 1.7 | 4.6 | 1.9 | 4.3 | 4.0 | 7.7 | 328 | 2530 |
| | BB | 0.062 | 0.36 | 0.23 | 0.53 | 0.51 | --- | --- | --- |
| CAS1 | AN | 0.13 | 1.3 | 0.38 | 2.0 | 0.52 | 9.2 | 454 | 4163 |
| | BB | <0.001 | 0.071 | 0.0017 | 0.066 | 0.0015 | --- | --- | --- |
| CAS2 | AN | 0.20 | 3.5 | 0.75 | 5.0 | 0.88 | 8.5 | 456 | 3888 |
| | BB | 0.0026 | 0.15 | 0.010 | 0.21 | 0.023 | --- | --- | --- |
| CAS3 | AN | 1.6 | 4.4 | 3.3 | 4.8 | 3.4 | 8.5 | 352 | 3000 |
| | BB | 0.082 | 0.30 | 0.49 | 0.30 | 0.23 | --- | --- | --- |
| CAS4 | AN | 1.3 | 4.1 | 1.9 | 4.5 | 1.7 | 8.1 | 372 | 3003 |
| | BB | 0.026 | 0.19 | 0.14 | 0.21 | 0.086 | --- | --- | --- |
| EAS1 | AN | 0.69 | 4.5 | 1.3 | 5.3 | 1.7 | 8.4 | 387 | 3251 |
| | BB | 0.021 | 0.32 | 0.050 | 0.47 | 0.12 | --- | --- | --- |
| EAS2 | AN | 4.4 | 17 | 6.7 | 19 | 9.2 | 8.8 | 344 | 3014 |
| | BB | 0.082 | 0.51 | 0.21 | 0.90 | 0.57 | --- | --- | --- |
| SAS | AN | 0.032 | 0.38 | 0.071 | 0.52 | 0.13 | --- | --- | --- |
| | BB | 0.0091 | 0.18 | 0.022 | 0.26 | 0.064 | --- | --- | --- |
| Others | AN | 0.59 | 2.0 | 0.87 | 2.2 | 1.1 | 7.7 | 402 | 3084 |
| | BB | 0.021 | 0.27 | 0.074 | 0.28 | 0.057 | --- | --- | --- |

[a] EUR: Europe, SIB: Siberia, GL: Greenland, NAM: North America, CAS: Central Asia, EAS: East Asia, SAS: Southeast Asia. These regions are defined in Fig. 1.

[b] AN: Anthropogenic (fossil fuel + biofuel), BB: Biomass burning.

[c] Values are shown only for regions/sources where their contributions are greater than 1%.