# Peer review of "Contrasting source contributions of Arctic black carbon to atmospheric concentrations, deposition flux, and atmospheric and snow radiative effects"

_Atmospheric Chemistry and Physics, 2021_

## Referee Comment (RC1)

In their manuscript "Contrasting source contributions of Arctic black carbon to atmospheric concentrations, deposition flux, and atmospheric and show radiative effects", the authors Matsui et al. evaluated the source attribution of Arctic BC concentration, deposition and radiative effects based on a global climate model CAM-ATRAS. The headline finding is that BC sources from high and low latitudes exhibit different contributions on Arctic BC loading, as well as its radiative effects. The manuscript is well organized, and the findings appear sound for the most part. The text could use some areas of clarification and potentially additional information. I believe the manuscript will be suitable for publication in *Atmospheric Chemistry and Physics* after the questions and issues below are addressed.

**GENERAL COMMENTS**

This study used three years of model simulations, but since some local events (e.g. Siberian biomass burning) can have a significant impact on Arctic BC, how representative are these three years, and can these three years be used to represent the Arctic climatology? It would be more helpful if the authors could assess the climatological representativeness of the simulated three years.

The methodological description needs additional work. Specifically, how many bins are used to represent fine particles (from 40 to 1250 nm)? How did you improve the model representation of activation processes in liquid clouds and removal processes in cumulus and mixed-phase clouds? There were 13 simulations, were they all nudged or free runs? How was the  $RE_{BC_TOA}$  estimated, was an additional run without the target source BC emission also made? The BC emission data was from GFEDv4, but did your model scale the emissions up as most models? How comparable are the modelled results and observations, are they all corrected for the standard temperature and pressure (STP)? Also, it seems that the decomposition of  $RE_{BC_TOA}$  in P8 seems should be included in the method. The calculation of HeightBC, FluxBC, and AlbedoBC also should be moved to the methods.

The model has been evaluated with observational data illustrating some of the uncertainties in the model. However, these uncertainties should be briefly evaluated to indicate whether and to what extent they may have an impact on the results.

**SPECIFIC COMMENTS**

P2 L49, "Unlike atmospheric BC, rain rate in the Arctic varies seasonally, with ...". This statement is inconsistent with the "Atmospheric BC mass ( $M_{BC}$ ) concentration in the Arctic show distinct seasonal variation" at the beginning of the paragraph. Please clarify.

P2 L59. Please define the radiative effect of BC first and be consistent with the IPCC definition, i.e. use RFari or ERFari.

P2 L74. Does the "light absorption efficiency" equal the  $MAC_{BC}$  used later in the manuscript? I would suggest that the terminology of this parameter be consistent throughout the article.

P3 L84. "We also show in this study that the light .....". This sentence is already your conclusion and should not appear in the introduction.

P3L94. Unclear how many bins in the fine particles.

P4 L138. "Figure S1 shows that the difference between ALL BC and the sum of BC tags is large where  $RE_{BC_SNOW}$  is large". Please clarify that you are comparing  $RE_{BC_SNOW}$  from ALL BC and BC tags. Also, this statement is not correct; many areas with low  $RE_{BC_SNOW}$  also exhibit large differences.

P4 L153. GL from the offline method is also much smaller than that from the online method.

P7 L285. "The largest contribution to Arctic BC" is not correct and inconsistent with the figure caption. Also, as the manuscript focuses on the Arctic, I suggest that Figure 9 be amended to show only the spatial distribution of the largest sources in the Arctic (not global distribution), so that it can be shown more clearly.

P8 L297. Contribution from North America is the largest, and the contribution of Siberia, Europe, and East Asia are the largest? Please clarify.

P8 L314. Should those values be annual mean values?

P8 L321-323. Sedlacek et al. showed that from observations, with the long-time aging, photolysis and other processes can decrease the coating thickness, mainly when the BB particles are transported in the free troposphere (https://agu.confex.com/agu/fm21/meetingapp.cgi/Paper/943236). This is inconsistent with the conclusion here that aging (23-30 days) can increase the coating thickness.

P10 L369. "Because  $M_{BC_DEP}$  is highest in summer, the contribution from biomass burning sources to  $M_{BC_DEP}$  is larger (16% from Siberia and 8.9% from North America 370 (>50°N)) than that to  $M_{BC_SRF}$  and  $M_{BC_COL}$ ". This is not clear; here should also mention that biomass burning aerosols in summer is also higher.

P10 L376. Please clarify at which wavelength was the MACBC caculated.

---

## Author Comment (AC1)

Response to reviewer #1

acp-2021-1091: "Contrasting source contributions of Arctic black carbon to atmospheric concentrations, deposition flux, and atmospheric and snow radiative effects" by H. Matsui et al.

We thank the reviewer very much for reading the paper carefully and giving us valuable comments. We revised the paper by taking into account the reviewers' comments. Considering reviewers' comments, we extended the model simulations to years 2009−2015. All figures and values in the manuscript have been revised, and a new paragraph and figures/table on inter-annual variability have been added. Main conclusions do not change by this revision. Detailed responses to individual comments and suggestions are given below.

Reviewer's comment:
This study used three years of model simulations, but since some local events (e.g. Siberian biomass burning) can have a significant impact on Arctic BC, how representative are these three years, and can these three years be used to represent the Arctic climatology? It would be more helpful if the authors could assess the climatological representativeness of the simulated three years.

Response:
Thank you for your useful comment. Considering this comment, we extended the model simulations and analysis to years 2009−2015 (7 years). We have revised all figures and statistics in the manuscript and added a paragraph for inter-annual variability of BC source contributions to the manuscript (Lines 317−328). As shown in Fig. 10 (also shown below as Fig. R1), inter-annual variability in BC source contributions is associated with inter-annual variability in BC emissions, mainly from biomass combustion sources. Qualitative characteristics of the major sources (e.g., which sources make large contributions) however do not change significantly between years. The main conclusions of this study (i.e., source contributions vary substantially among the five BC variables) are found for all years simulated.

In the revised manuscript, the new paragraph for inter-annual variability of BC source contributions is described as follows: "*The source contributions of BC show year-to-year variability, mainly in response to interannual variations in BC emissions at mid- and high latitudes (Fig. 10). For the years 2012, 2015, and 2016, BC emissions from*

*biomass burning sources north of 50°N are about twice those for the other years, and the contributions from biomass burning sources to $M_{BC\_COL}$ and $RE_{BC\_TOA}$ are larger in the Arctic (Figs 10b and 10d). The contributions from biomass burning sources in Siberia and North America (>50°N) to $M_{BC\_DEP}$, $RE_{BC\_TOA}$, and $RE_{BC\_SNOW}$ vary between years by a factor of 3.4 to 6.4 (by up to about 20%), with large interannual variability (Fig. 10, Table S1). Compared with those of biomass burning BC, the source contributions of anthropogenic BC show smaller interannual variability: source contributions generally vary within a factor of 2 (within 10%). Our anthropogenic BC emissions north of 50°N decrease by about 10% from 2009 to 2015 (Fig. S6a). In addition, the atmospheric lifetime of anthropogenic BC north of 50°N is longest in 2009 (Fig. S6b). For these reasons, the source contribution of anthropogenic BC is largest in 2009 and tends to decrease in subsequent years (Fig. 10). Overall, the source contributions to the five BC variables show interannual variation to some extent, but the qualitative source characteristics (e.g., which sources make large contributions) do not change significantly during the simulation periods.*" (Lines 317−328).

[Figure]

Figure R1: Year-to-year variations of annual-mean source contributions to (a) $M_{BC\_SRF}$, (b) $M_{BC\_COL}$, (c) $M_{BC\_DEP}$, (d) $RE_{BC\_TOA}$, and (e) $RE_{BC\_SNOW}$ in the Arctic for years from 2009 to 2015 (left axis). The filled and shaded areas indicate contributions from anthropogenic and biomass burning sources, respectively. The black and grey lines show BC concentrations, deposition flux, or radiative effects from all (anthropogenic

+ biomass burning) sources and anthropogenic sources, respectively (right axis).

Reviewer's comment:

The methodological description needs additional work. Specifically, how many bins are used to represent fine particles (from 40 to 1250 nm)?

Response:

Five particle size bins are used to represent fine particles (40−1250 nm). Eight BC mixing state bins are used for each of these five bins. Therefore, there are 40 bins (5 size bins × 8 mixing state bins) for fine particles in our model (Matsui and Mahowald, 2017). We have added this information to the manuscript (Line 93).

Reviewer's comment:

How did you improve the model representation of activation processes in liquid clouds and removal processes in cumulus and mixed-phase clouds?

Response:

In Liu and Matsui (2021b), we separately represented activated and non-activated aerosols in convective clouds and introduced gradual activation processes of aerosols during upward transport. This representation allows consistent calculations of the transport, activation, and removal processes of aerosols in convective clouds. We also introduced the reduction in precipitation removal efficiency of aerosols in mixed-phase clouds by the Wegener-Bergeron-Findeisen process (Liu and Matsui, 2021b), and following Cozic et al. (2007), we represented precipitation removal efficiency as a function of the ice mass fraction in mixed-phase clouds. We have added these descriptions to the revised manuscript (Lines 111−116).

Reviewer's comment:

There were 13 simulations, were they all nudged or free runs?

Response:

All simulations in this study were nudged by the Modern-Era Retrospective analysis for Research and Applications version 2 (MERRA2) for wind speed and direction and temperature in the free troposphere (<800 hPa). We have added this sentence to the revised manuscript (Line 165−166).

Reviewer's comment:

How was the $RE_{BC\_TOA}$ estimated, was an additional run without the target source BC emission also made?

Response:

Three radiative transfer calculations (considering ALL BC, excluding anthropogenic BC in the target area from ALL BC, and excluding biomass burning BC in the target area from ALL BC) were performed for each simulation to estimate instantaneous BC radiative effects from the target source. We have added this sentence to the manuscript (Lines 136−138).

Reviewer's comment:

The BC emission data was from GFEDv4, but did your model scale the emissions up as most models?

Response:

No scaling was done in this study. Some recent studies have suggested that biomass burning emissions are underestimated (e.g., Reddington et al., 2016, Mallet et al., 2021), but their scaling factors have large uncertainties. In this study, we used the GFED data directly. We have clarified this point in the revised manuscript (Lines 169−170).

Reviewer's comment:

How comparable are the modelled results and observations, are they all corrected for the standard temperature and pressure (STP)?

Response:

BC concentrations are shown at STP in both observations and model simulations. We have clarified this point in the revised manuscript (figure caption in Figure 2; Lines 720−721).

Reviewer's comment:

Also, it seems that the decomposition of $RE_{BC\_TOA}$ in P8 seems should be included in the method. The calculation of $Height_{BC}$, $Flux_{BC}$, and $Albedo_{BC}$ also should be moved to the methods.

Response:

        The equations for $RE_{BC\_TOA}$, $Height_{BC}$, $Flux_{BC}$, and $Albedo_{BC}$ (equations 2−5) are used only in section 3.4. We think it is better to explain these equations within this section.

        The model has been evaluated with observational data illustrating some of the uncertainties in the model. However, these uncertainties should be briefly evaluated to indicate whether and to what extent they may have an impact on the results.

Response:

        There are uncertainties in comparisons between observations and model simulations. For example, observation data (e.g., aircraft and snow BC data) and model simulation outputs have different spatial and temporal scales. Observed data are for a specific location and time, with time scales of minutes (aircraft observations) to days (snow observations), whereas in comparisons with aircraft observations, we used monthly model outputs for a specific region (e.g., 60−80°N and 140−170°W for HIPPO) and in comparisons with snow BC, we used monthly averaged model outputs over a horizontal grid of about 200 km. Observations suggest that snow BC concentrations vary widely over fine spatial and temporal scales, but model outputs do not fully resolve this variability (Fig. 5a). These uncertainties in comparisons between observations and models are seen not only in this study but in all studies using both observations and model simulations (e.g., Schutgens et al., 2017). Despite these uncertainties in observation-model comparisons, the results obtained in this study are comparable to or better than those obtained by previous studies in terms of the reproducibility of BC observations in the Arctic. These discussions are added to the revised manuscript (Lines 227−236).

        There are also uncertainties in the representation of aerosols and their processes. We have focused on these uncertainties in our recent studies (e.g., Matsui and Moteki, 2020; Liu and Matsui, 2021; Matsui and Liu, 2021). However, we think evaluating their impacts on BC source contributions require many additional simulations and is beyond the range of what can be done in a single study. We will evaluate the uncertainties in aerosols and their processes in a future study.

        P2 L49, "Unlike atmospheric BC, rain rate in the Arctic varies seasonally, with …". This statement is inconsistent with the "Atmospheric BC mass ($M_{BC}$) concentration

in the Arctic show distinct seasonal variation" at the beginning of the paragraph. Please clarify.

Response:
It is reasonable that atmospheric concentrations are higher during low precipitation seasons (winter and spring) and lower during high precipitation seasons (summer).

Reviewer's comment:
P2 L59. Please define the radiative effect of BC first and be consistent with the IPCC definition, i.e. use RFari or ERFari.

Response:
Because this part is in Introduction, detailed definition is not given in this part. RE is defined in the Methods (Lines 135−138).

For all BC variables in this study, we estimate the source contributions of all BC (both anthropogenic and biomass burning BC) in the present-day (PD) climate. RE is defined as the difference in the instantaneous radiative balance with and without BC in PD (Lines 135−138). On the other hand, RFari and ERFari are estimated from the difference between PD and preindustrial (PI) simulations, and the contribution from only anthropogenic BC is considered if the change in biomass burning emissions from PI to PD is small. RE is used in this study to compare the source contributions of radiative effects, mass concentrations, and deposition flux consistently (for both anthropogenic and biomass burning BC in the PD climate). RFari is used in part for comparisons with previous studies (Lines 246−251).

Reviewer's comment:
P2 L74. Does the "light absorption efficiency" equal the $MAC_{BC}$ used later in the manuscript? I would suggest that the terminology of this parameter be consistent throughout the article.

Response:
We have deleted "light absorption efficiency" from this sentence.

Reviewer's comment:
P3 L84. "We also show in this study that the light …..". This sentence is already

your conclusion and should not appear in the introduction.

Response:

We do not remove this sentence because it briefly explains what we are showing in this manuscript and we believe it is useful to readers.

Reviewer's comment:

P3L94. Unclear how many bins in the fine particles.

Response:

Please see the response to the first comment.

Reviewer's comment:

P4 L138. "Figure S1 shows that the difference between ALL BC and the sum of BC tags is large where $RE_{BC\_SNOW}$ is large". Please clarify that you are comparing $RE_{BC\_SNOW}$ from ALL BC and BC tags. Also, this statement is not correct; many areas with low $RE_{BC\_SNOW}$ also exhibit large differences.

Response:

We have removed this sentence from the manuscript.

"you are comparing $RE_{BC\_SNOW}$ from ALL BC and BC tags": this is clearly indicated both in the text (Lines 140−144) and in the figure caption of Fig. S1.

Reviewer's comment:

P4 L153. GL from the offline method is also much smaller than that from the online method.

Response:

This figure (Figure 7) has been revised in this revision. The contribution from GL is small even in the Online calculation.

Reviewer's comment:

P7 L285. "The largest contribution to Arctic BC" is not correct and inconsistent with the figure caption. Also, as the manuscript focuses on the Arctic, I suggest that Figure 9 be amended to show only the spatial distribution of the largest sources in the Arctic (not global distribution), so that it can be shown more clearly.

Response:

The word "Arctic" has been removed (Line 33). As the information near source regions is also important (Line 305), we use the current figure instead of the Arctic-centered figure. The current figure clearly shows the source regions with the largest contributions in the Arctic.

Response:

We have revised this sentence as follows: "*For $RE_{BC\_SNOW}$, which is limited to land areas, North America's (>50°N) contribution is the largest over 53% of the Arctic area (over the North American side of the Arctic), and the contributions of Siberia, Europe, and East Asia are the largest over 40%, 4.5%, and 3.2%, respectively, of the Arctic area (over the Siberian side of the Arctic) (Fig. 9e).*" (Lines 313−316).

Response:

We have clarified that these values are annual mean values (Line 344).

Response:

The results shown by the reviewer are not consistent with our aircraft observations. In Ohata et al. (2021a), we have shown that thickly-coated BC particles are abundant (both in the boundary layer and in the free troposphere) in the Arctic during spring (Fig. 8b in Ohata et al. (2021a)). The SP2 results of ARCTAS aircraft observations

(spring and summer 2008) also indicate the existence of many thickly-coated BC particles in the Arctic. The results of this study are consistent with these aircraft observations. The reviewer gave us an abstract of the AGU meeting, but we cannot determine the importance of photolysis or other processes from this information alone.

Reviewer's comment:

P10 L369. "Because $M_{BC\_DEP}$ is highest in summer, the contribution from biomass burning sources to $M_{BC\_DEP}$ is larger (16% from Siberia and 8.9% from North America (>50°N)) than that to $M_{BC\_SRF}$ and $M_{BC\_COL}$". This is not clear; here should also mention that biomass burning aerosols in summer is also higher.

Response:

We have added the following sentence to the manuscript: "*The contributions of biomass burning sources to $M_{BC\_SRF}$, $M_{BC\_COL}$, and $M_{BC\_DEP}$ are larger during summer months.*" (Lines 398−399).

Reviewer's comment:

P10 L376. Please clarify at which wavelength was the $MAC_{BC}$ calculated.

Response:

We have added the words "at the wavelength of 550 nm" to this sentence (Line 408).

References:
Cozic, J., et al.: Scavenging of black carbon in mixed phase clouds at the high alpine site Jungfraujoch, Atmos. Chem. Phys., 7, 1797–1807, doi:10.5194/acp-7-1797-2007, 2007.
Liu, M. and Matsui, H.: Improved simulations of global black carbon distributions by modifying wet scavenging processes in convective and mixed-phase clouds, J. Geophys. Res. Atmos., 126, e2020JD033890, doi:10.1029/2020JD033890, 2021.
Mallet, M., et al.: Climate models generally underrepresent the warming by Central Africa biomass-burning aerosols over the Southeast Atlantic, Sci. Adv., 7, eabg9998, doi:10.1126/sciadv.abg9998, 2021.
Matsui, H. and Liu, M.: Importance of supersaturation in Arctic black carbon simulations, J. Climate, 34, 7843−7856, doi:10.1175/JCLI-D-20-0994.1, 2021.
Matsui, H. and N. Mahowald: Development of a global aerosol model using a two-dimensional sectional method: 2. Evaluation and sensitivity simulations, J. Adv. Model.

Earth Syst., 9, 1887–1920, doi:10.1002/2017MS000937, 2017.

Matsui, H. and Moteki, N.: High sensitivity of Arctic black carbon radiative effects to subgrid vertical velocity in aerosol activation, Geophys. Res. Lett., 47, e2020GL088978, doi:10.1029/2020GL088978, 2020.

Ohata, S., et al.: Arctic black carbon during PAMARCMiP 2018 and previous aircraft experiments in spring, Atmos. Chem. Phys., 21, 15861−15881, doi:10.5194/acp-21-15861-2021, 2021a.

Reddington, C. L., et al.: Analysis of particulate emissions from tropical biomass burning using a global aerosol model and long-term surface observations, Atmos. Chem. Phys., 16, 11083-11106, doi:10.5194/acp-16-11083-2016, 2016.

Schutgens, N., et al.: On the spatio-temporal representativeness of observations, Atmos. Chem. Phys., 17, 9761–9780, doi:10.5194/acp-17-9761-2017, 2017.

Response to reviewer #2

acp-2021-1091: "Contrasting source contributions of Arctic black carbon to atmospheric concentrations, deposition flux, and atmospheric and snow radiative effects" by H. Matsui et al.

We thank the reviewer very much for reading the paper carefully and giving us valuable comments. We revised the paper by taking into account the reviewers' comments. Considering reviewers' comments, we extended the model simulations to years 2009−2015. All figures and values in the manuscript have been revised, and a new paragraph and figures/table on inter-annual variability have been added. Main conclusions do not change by this revision. Detailed responses to individual comments and suggestions are given below.

Reviewer's comment:
1) My one major concern with the entire paper and analysis is the reliance on three years only, to represent a climatology. There is significant interannual variability in BC emissions, transport, loading, precipitation etc., which is not touched on in the analysis but which is crucial for understanding the observed conditions in the Arctic - and for a realistic model representation. I would urge the authors to either document whether the three years they have used really can be said to represent a climatology (e.g. using extended simulations, or, if this is not practical, longer time series from other models that are already available through AeroCom, CMIP6 or similar), or - preferably - to add discussion of the interannual variability in their results throughout. This would be a major addition, of course, but it would also markedly strengthen the conclusions and community relevance of the paper.

Response:
Considering this reviewer's important comment, model simulations have been extended to years 2008−2015 in this revision. We have made analysis for years 2009−2015 (7 years) and revised all figures and statistics in the manuscript. We have also added a paragraph for inter-annual variability of BC source contributions to the manuscript (Lines 317−328). As shown in Fig. 10 (also shown below as Fig. R1), inter-annual variability in BC source contributions is associated with inter-annual variability in BC emissions, mainly from biomass combustion sources. We have clarified that the qualitative source characteristics (e.g., which sources make large contributions) do not

change significantly during the simulation periods whereas the quantitative values of source contributions vary to some extent interannually.

The new paragraph for inter-annual variability of BC source contributions is described as follows: "*The source contributions of BC show year-to-year variability, mainly in response to interannual variations in BC emissions at mid- and high latitudes (Fig. 10). For the years 2012, 2015, and 2016, BC emissions from biomass burning sources north of 50°N are about twice those for the other years, and the contributions from biomass burning sources to $M_{BC\_COL}$ and $RE_{BC\_TOA}$ are larger in the Arctic (Figs 10b and 10d). The contributions from biomass burning sources in Siberia and North America (>50°N) to $M_{BC\_DEP}$, $RE_{BC\_TOA}$, and $RE_{BC\_SNOW}$ vary between years by a factor of 3.4 to 6.4 (by up to about 20%), with large interannual variability (Fig. 10, Table S1). Compared with those of biomass burning BC, the source contributions of anthropogenic BC show smaller interannual variability: source contributions generally vary within a factor of 2 (within 10%). Our anthropogenic BC emissions north of 50°N decrease by about 10% from 2009 to 2015 (Fig. S6a). In addition, the atmospheric lifetime of anthropogenic BC north of 50°N is longest in 2009 (Fig. S6b). For these reasons, the source contribution of anthropogenic BC is largest in 2009 and tends to decrease in subsequent years (Fig. 10). Overall, the source contributions to the five BC variables show interannual variation to some extent, but the qualitative source characteristics (e.g., which sources make large contributions) do not change significantly during the simulation periods.*" (Lines 317−328).

[Figure]

Figure R1: Year-to-year variations of annual-mean source contributions to (a) $M_{BC\_SRF}$, (b) $M_{BC\_COL}$, (c) $M_{BC\_DEP}$, (d) $RE_{BC\_TOA}$, and (e) $RE_{BC\_SNOW}$ in the Arctic for years from 2009 to 2015 (left axis). The filled and shaded areas indicate contributions from anthropogenic and biomass burning sources, respectively. The black and grey lines show BC concentrations, deposition flux, or radiative effects from all (anthropogenic + biomass burning) sources and anthropogenic sources, respectively (right axis).

Reviewer's comment:

2) In the description of the simulations, I could not find the model setup. I assume you are running with nudged simulations for the years 2009-2011? (If not, the RF calculations presented later would not be correct, so I hope this is the case.) I recommend documenting this is some more detail.

Response:

The meteorological data of the Modern-Era Retrospective analysis for Research and Applications version 2 (MERRA2) were used for nudging of wind speed and direction and temperature in the free troposphere (<800 hPa). We have clarified this point in the revised manuscript (Line 165−166).

Reviewer's comment:

3) The global mean lifetime of BC in the baseline model is given as 5.6 days.

This is at the upper end of recent estimates (see e.g. Lund et al. 2018 (https://www.nature.com/articles/s41612-018-0040-x), and could be expected to affect the transport of Asian BC into the Arctic. (Or rather, the processes that lead to this lifetime indicate that ageing and wet removal are slow enough to allow for transport into the Arctic.) However, the modelled lifetime, and therefore the type of results shown in this study, are very sensitive to how these processes are parameterized. There are currently no sensitivity studies of this in the manuscript. Would it be worth the effort to check how sensitive the results are to a realistic change in wet removal/ageing? If this dramatically changes the source region composition, then that is of course of high interest to the community as it will indicate a major source of model diversity in Arctic BC RF.

Response:

Previous studies have estimated short BC lifetimes to make consistency with aircraft observations of remote areas such as HIPPO and ATom. On the other hand, our recent study (Liu and Matsui, 2021) has shown that improved removal processes in convective and mixed-phase clouds can greatly improve the consistency with BC observations both in the upper troposphere in the tropics and in middle and lower troposphere in the Arctic, even without shortening the BC lifetimes. Liu and Matsui (2021) discussed these results by citing Lund et al. (2018).

As the reviewer pointed out, there are large uncertainties in the treatment of BC aging, activation, and removal processes in models. We have focused on these uncertainties in our recent studies (e.g., Matsui and Moteki, 2020; Liu and Matsui, 2021; Matsui and Liu, 2021). However, we think that evaluating the effects of uncertainties in these processes on source contributions requires many additional simulations and it is beyond the range of a single study. We will discuss the importance of these uncertainties in more detail in a future study.

Reviewer's comment:
Figure 5: This is not a major point of the paper, but it seems to me that the model has essentially no interannual variability in BC on /in snow. There is a geographical variation, but for each location the model points all lie on a virtually straight line while the observations range over 1-2 orders of magnitude. This is perhaps worth mentioning? See also my first point above.

Response:

In this revision, this figure was revised so that the year/month of observations

and model simulations are consistent, but the variability of snow BC concentrations in observations is still significantly larger than that in model simulations. One of the main reasons for the larger variability of observed snow BC concentrations is different spatial and temporal scales of the observed and simulated data. Observed data are for a specific location and time, with time scales of the order of days. Model simulations, on the other hand, use monthly averaged outputs over a horizontal grid of about 200 km. Observations suggest that snow BC concentrations vary widely over fine spatial and temporal scales, but model outputs do not fully resolve this variability. We have added these discussions to the manuscript (Lines 227−236).

Figure 6: The purple regions are not easy to interpret. Is this the lowest color in the scale? (It seems so, but I had to zoom in on the colorbar on a large screen to see it.)

Response:

We have added the following sentence to the figure caption of Fig. 6: "*Purple shows areas where values are below the minimum shown on the colour bars.*" (Lines 756−757).

Reviewer's comment:

Line 285: "largest contributions to Arctic BC" -> this should be just "BC" I think. The figure shows the dominating source regions for the entire NH, not just the Arctic.

Response:

We have changed from "Arctic BC" to "BC", as the reviewer suggested (Line 303).

Reviewer's comment:

Line 312: AeroCom models -> AeroCom Phase II models (the RF range will differ for the various AeroCom phases)

Response:

We have added "Phase II" to this sentence, as the reviewer suggested (Line 343).

References:
Liu, M. and Matsui, H.: Improved simulations of global black carbon distributions by

modifying wet scavenging processes in convective and mixed-phase clouds, J. Geophys. Res. Atmos., 126, e2020JD033890, doi:10.1029/2020JD033890, 2021.

Matsui, H. and Liu, M.: Importance of supersaturation in Arctic black carbon simulations, J. Climate, 34, 7843−7856, doi:10.1175/JCLI-D-20-0994.1, 2021.

Matsui, H. and Moteki, N.: High sensitivity of Arctic black carbon radiative effects to subgrid vertical velocity in aerosol activation, Geophys. Res. Lett., 47, e2020GL088978, doi:10.1029/2020GL088978, 2020.

---

## Author Response (AR2)

Response to Editor

acp-2021-1091: "Contrasting source contributions of Arctic black carbon to atmospheric concentrations, deposition flux, and atmospheric and snow radiative effects" by H. Matsui et al.

We thank the Editor very much for reading the paper carefully and giving us valuable comments. We revised the paper by taking into account all the Editor's comments. Detailed responses to individual comments and suggestions are given below.

Editor's comment:

Having said that, I also agree with the assessment that the explanation of the enhanced mass absorption coefficient for Asian BC would still benefit from a bit more scrutiny and improved explanation.

Response:

As the reviewer pointed out, the large MAC values of anthropogenic BC from Asia in our model simulations may be because of fast aging processes near the source regions. Considering this point, we have revised the manuscript as follows: "*Compared with anthropogenic BC from Siberia, Europe, and North America (>50°N), anthropogenic BC particles from Asia have a higher fraction of thickly coated BC particles (which have higher $MAC_{BC}$) and a lower fraction of thinly coated BC particles (which have lower $MAC_{BC}$) (Fig. 12a). The higher fraction of thickly coated BC from Asia might be explained by fast aging processes near their sources, where the concentrations of condensable gases emitted with BC are high, and by the higher fraction of anthropogenic BC from Asia in the upper troposphere in the Arctic (Fig. S7) and its longer lifetime in the Arctic (24−30 days) (Fig. 12b).*" (Lines 365-371).

In addition, many sources may contribute to the observed thickly coated BC in the Arctic region, as in the reviewer's comment. Considering this point, we have added the following discussion to the revised manuscript: "*The fraction of thickly coated BC was observed to be high in the Arctic in recent aircraft measurements by Ohata et al. (2021b), consistent with our model simulations, although it is difficult to observe the dependence of BC mixing states on emission sources. Our simulation results indicate the importance of understanding the differences in BC mixing states among sources and the mechanisms that control them in evaluating the source contribution of BC to $RE_{BC\_TOA}$ in the Arctic.*" (Lines 374-378).

Editor's comment:

An understanding of this relies on the description of the aerosol microphysics. However, in the present form, insufficient detail is provided to understand key processes, in particular regarding the mixing state and the radiative properties:

- It is not entirely clear from the description, if all microphysical processes act on the full 2D bins or if they are estimated from simulating microphysical processes on the 17 size bins and subsequent allocation to mixing state bins.

Response:

Microphysical processes are calculated for the full 2D bins (47 bins). Changes in particle size and mixing state by microphysical processes are calculated for all the 47 bins, and bin shifting by these processes is calculated by a two moment (mass and number) advection scheme (Simmel and Wurzler, 2006) for particle size bins and the moving center approach (Jacobson, 1997) for mixing state bins (Matsui, 2017). We have added these sentences to the revised manuscript (Lines 98-105).

Editor's comment:

- The calculation of radiative properties as "calculated theoretically (Bohren and Huffman, 1998;" is not sufficient. What theories are used and what are the actual references (noting that Bohren and Huffman is a full textbook not a specific reference)? How is mixing represented (effective medium approximations, core/shell,...) and what are the related uncertainties?

Response:

For optical properties, we assumed the core/shell treatment for internally mixed BC in the fine particle size bins (40−1250 nm in diameter) and the well-mixed treatment for the other particles (for pure BC and BC-free particles in the fine particle size bins and for all particles larger than 1250 nm or smaller than 40 nm) (Matsui, 2017). CAM-ATRAS uses look-up tables of optical parameters (extinction coefficient, single scattering albedo, and asymmetry factor) calculated based on the codes for homogeneous and coated spheres (Appendices A and B in Bohren and Huffman (1998)). The core/shell treatment could underestimate the mass absorption cross section of BC ($MAC_{BC}$) for large particles (Forestieri et al., 2018), but as shown by Matsui et al. (2018b), the enhancement of BC absorption by the core/shell treatment is comparable to that by other mixing state assumptions such as the dynamic effective medium approximation (Chylek et al., 1984;

Jacobson, 2006) and the Bruggeman mixing rule (Jacobson, 2006). We have clarified these points in the revised manuscript (Lines 108-116).

Editor's comment:
       - And finally, the representation of the mixing state in microphysical models tends to be heavily affected by assumptions made during emissions (which often dominate over the actual microphysical processes). While some BC sources emit fairly pure particles, other sources have already a high degree of internal mixing. How is the mixing state dealt with at point of emission (or of formation of secondary organics)?

Response:
       Given the large uncertainty in the assumption of BC mixing states in emissions (Matsui, 2020), we assumed BC is emitted as pure BC and the other species as BC-free particles (Matsui et al., 2018b). In reality, the mixing state of emitted BC particles depends on the types of sources. Matsui et al. (2018b) made a simulation assuming that 50% of BC mass is emitted as pure BC and the other 50% of BC as internally mixed BC with the shell (organic aerosol) to core (BC) diameter ratio of 1.1 for fossil fuel sources and 1.4 for biofuel and biomass burning sources and showed that global mean $RE_{BC\_TOA}$ in this simulation is about 10% larger than that in the simulation assuming pure BC for all BC emissions. We have clarified these points in the revised manuscript (Lines 186-191).

Response to reviewer #1

acp-2021-1091: "Contrasting source contributions of Arctic black carbon to atmospheric concentrations, deposition flux, and atmospheric and snow radiative effects" by H. Matsui et al.

We thank the reviewer very much for reading the paper carefully and giving us valuable comments. We revised the paper by taking into account the reviewers' comments.

Reviewer's comment:
The authors calculated a larger MAC for the BC from the Asian region, suggesting that it is due to its 30-day long transport and ageing process that causes the thickest coating of its BC.
Here I would argue that even if the coating of BC from Asian sources is thick, it does not necessarily mean that it is due to ageing and mixing in transport. It is also likely that the high concentration of condensable gas from Asian emissions would result in a thicker coating a few hours after BC is emitted.
Furthermore, the authors argue that the thicker BC coating observed in the Arctic shows the contribution of aerosol ageing in transport. However, this does not indicate that this observed thick-coated BC is of Asian origin. Other sources with shorter transport times can also contribute to thickly coated BC particles. After too long of a transport time, for example 30 days as estimated by the authors, coating thickness may decrease with photolysis. The authors cannot prove that coating is consistently increasing or not decreasing over such a long transport time. While this may be a common problem in the current models, I would propose to add a few sentences of discussion to clarify this possible uncertainty.

Response:
As the reviewer pointed out, the large MAC values of anthropogenic BC from Asia in our model simulations may be because of fast aging processes near the source regions. Considering this point, we have revised the manuscript as follows: "*Compared with anthropogenic BC from Siberia, Europe, and North America (>50°N), anthropogenic BC particles from Asia have a higher fraction of thickly coated BC particles (which have higher $MAC_{BC}$) and a lower fraction of thinly coated BC particles (which have lower $MAC_{BC}$) (Fig. 12a). The higher fraction of thickly coated BC from Asia might be explained by fast aging processes near their sources, where the concentrations*

*of condensable gases emitted with BC are high, and by the higher fraction of anthropogenic BC from Asia in the upper troposphere in the Arctic (Fig. S7) and its longer lifetime in the Arctic (24−30 days) (Fig. 12b).*" (Lines 365-371).

In addition, many sources may contribute to the observed thickly coated BC in the Arctic region, as in the reviewer's comment. Considering this point, we have added the following discussion to the revised manuscript: "*The fraction of thickly coated BC was observed to be high in the Arctic in recent aircraft measurements by Ohata et al. (2021b), consistent with our model simulations, although it is difficult to observe the dependence of BC mixing states on emission sources. Our simulation results indicate the importance of understanding the differences in BC mixing states among sources and the mechanisms that control them in evaluating the source contribution of BC to $RE_{BC\_TOA}$ in the Arctic.*" (Lines 374-378).